# Applying machine learning to predict stunting in children under 5 years old based on water, sanitation and hygiene behaviors and infrastructure

**Sanaya Sinharoy**[1], **Heather Reese**[2], **Thomas Clasen**[2], **Sheela S. Sinharoy**[3]*

**1** Independent Researcher, Doylestown, Pennsylvania, United States of America, **2** Gangarosa Department of Environmental Health, Rollins School of Public Health, Emory University, Atlanta, Georgia, United States of America, **3** Hubert Department of Global Health, Rollins School of Public Health, Emory University, Atlanta, Georgia, United States of America

* sssinh2@emory.edu

## Abstract

### Objective

Child stunting continues to pose a substantial global health challenge, requiring multifaceted strategies that combine conventional epidemiological approaches with advanced analytic methods. The aim of this study was to determine the most effective machine learning model for predicting stunting based on water, sanitation, and hygiene behaviors and infrastructure, with the goal of identifying high-risk children who would benefit most from targeted interventions.

### Methods

This study was a secondary analysis of data from a matched cohort study assessing the effectiveness of combined on-premise piped water and improved sanitation for improved health outcomes in rural Odisha, India. Data for the parent study were collected from 2,398 households with a child under five years of age across 90 villages, and complete data were available for 1,196 children. Feature engineering techniques were employed to identify the most relevant predictors and utilized structural equation modeling, forward selection, backward elimination, and least absolute shrinkage and selection operator techniques. Five machine learning algorithms commonly used for binary classification tasks were compared: logistic regression, classification tree, support vector machine, neural network, and extreme gradient boosting.

### Results

Among 1,196 children analyzed, the extreme gradient boosting model with forward selection feature engineering best predicted stunting based on water, sanitation, and hygiene (WaSH) factors. It correctly identified 81% of stunted children and 92% of

**Data availability statement:** The data underlying the findings described in this manuscript have been uploaded to FigShare. They can be accessed via the following private link, which will be updated to the below publicly available DOI upon acceptance: 10.6084/m9.figshare.28711070.

**Funding:** This work was supported by the Bill & Melinda Gates Foundation [grant numbers OPP1008048 and OOP1125067]. The funder had no role in the study design, data collection, analysis, preparation of the manuscript, or decision to publish. There was no additional external funding received for this study.

**Competing interests:** The authors have declared that no competing interests exist.

non-stunted children, with an overall accuracy of 88%. The model's area under the receiver operating characteristic curve (AUROC) was 0.959 (95% CI: 0.949–0.968), indicating that WaSH factors strongly predict child stunting when analyzed using this advanced machine learning technique. Four WaSH factors were identified as having the strongest power to predict stunting in our sample: improved sanitation coverage, presence of a handwashing station, piped water coverage, and availability of preferred drinking water source.

## Conclusions

The results demonstrate the efficacy of machine learning algorithms, especially extreme gradient boosting to potentially inform targeted WaSH interventions for reducing childhood stunting in resource-limited settings. However, these findings require external validation in other populations, and the complete-case analysis approach (excluding 35% of children with missing data) may limit generalizability to settings with less systematic data collection.

## Introduction

Child stunting, defined as height-for-age more than two standard deviations below the World Health Organization Child Growth Standards reference median for age and sex, remains a substantial global health challenge, particularly in low- and middle-income countries [1]. Stunting is a complex, multifactorial condition that reflects the cumulative effects of inadequate nutrition, repeated infections, and poor environmental conditions during the critical period of early childhood development [2–4]. The adverse outcomes associated with stunting extend beyond childhood, impacting adult health, educational attainment, and economic productivity [2,3,5].

Water, sanitation, and hygiene (WaSH) interventions have been identified as having potential to improve child nutritional status as measured through linear growth [6]. Poor WaSH conditions increase the risk of enteric infections and environmental enteric dysfunction, leading to impaired nutrient absorption and utilization, which in turn contributes to linear growth faltering [7,8]. However, the evidence linking WaSH interventions to improvements in child linear growth has been mixed, with some studies reporting significant associations [9–12] and others finding no effect [13–16].

The complex interplay between WaSH and stunting suggests that the application of advanced analytical methods could improve our understanding of these relationships and help develop tools for identifying high-risk children who may benefit from targeted public health programs. Machine learning algorithms have emerged as powerful tools for predicting health outcomes and identifying important risk factors from large, complex datasets [17–19], including child stunting [20–22]. However, it is important to note that machine learning models identify predictive associations rather than causal relationships; they can predict which children are at highest risk based on observed patterns, but do not establish that interventions targeting predictors will necessarily improve outcomes.

This study focuses on the state of Odisha, India, where child stunting, while lower than the national average, remains a concern with a prevalence of 31% among children under five, compared to the national average of 36% [23]. Overall, child stunting in India persists alongside challenges in WaSH practices, with 19% of households nationally still practicing open defecation, rising to 26% in rural areas [23]. Notably, Odisha has one of the lowest rates of toilet access in the country, with only 71% of households having toilet facilities [23]. The coexistence of high stunting rates and poor sanitation, especially in rural areas, provides a strong rationale for examining the relationship between WaSH practices and child stunting in India, potentially developing predictive tools to identify high-risk children in geographies like rural Odisha.

This study applies machine learning techniques to predict childhood stunting based on WaSH behaviors and infrastructure, addressing limitations of our previous structural equation modeling (SEM) approach [24]. While the previous approach effectively identified pathways between WaSH factors and height-for-age *z* scores, it has specific constraints: it primarily models linear relationships, requiring strong theoretical assumptions about causal structures.

Our machine learning approach offers certain advantages for risk prediction: (1) algorithms like XGBoost can capture complex non-linear relationships and interactions between WaSH variables without requiring pre-specified structural assumptions; (2) it quantifies predictive performance with metrics directly relevant to field applications such as sensitivity (recall) and specificity; (3) it identifies the combination of WaSH factors that maximizes prediction accuracy rather than focusing on individual pathways; and (4) it provides a framework for developing practical screening tools that could identify high-risk children in resource-limited settings. Importantly, while our previous SEM analysis provided insights into causal mechanisms (how WaSH factors influence growth through specific pathways), the machine learning approach developed here focuses on prediction (identifying which children are at highest risk based on WaSH and demographic characteristics). These predictive models can inform programmatic decisions about resource allocation and screening priorities, but do not replace the need for rigorous causal inference methods when evaluating intervention effectiveness. By systematically comparing multiple algorithms and feature engineering techniques, this study transforms insights about WaSH-stunting relationships into actionable tools for risk identification and targeted program planning.

## Methods

### Study design

This study used data from an original primary research initiative focused on evaluating the effectiveness of a household-level combined water and sanitation intervention for improved health outcomes in rural areas of Odisha, India, specifically within the Ganjam and Gajapati districts [10,25]. The water and sanitation interventions were implemented by Gram Vikas, a non-governmental organization based in Odisha, and included household pour-flush toilets with dual soak-away pits, attached bathing rooms, and piped water connections with taps in the toilets, bathing areas, and kitchens. For the evaluation study, 45 villages were randomly selected from a list of all intervention villages, and 45 control villages were matched to the intervention villages through a process of restriction, matching, and exclusion. Enrollment eligibility included households with children under five years of age, with up to 40 households enrolled per village. A total of 2398 households were enrolled in the primary study. Details regarding the intervention and original primary research study design have been previously documented [10,25].

### Ethics

Informed written consent was obtained from the male and/or female heads of all enrolled households for participation in the original study, including collection of household survey data and child anthropometric measurements. Ethical approval was granted by the Ethics Committee of the London School of Hygiene and Tropical Medicine, U.K (No. 9071), and the Institute Ethics Committee of the Kalinga Institute of Medical Sciences at KIIT University, Bhubaneswar, India (KIMS/KIIT/IEC/053/2015). All personal identifiers were removed from the dataset before transfer. Anonymized data were

provided to Emory University under a data transfer agreement and analysis was approved by the Emory University IRB (IRB00079717). The machine learning analysis presented here used deidentified secondary data, for which additional human subjects ethics approval was not required.

## Data source

Data from the original primary research initiative were collected over four rounds from 1st June 2015–31st October 2016, and child anthropometric data utilized in our analysis were collected in a single round between February to June 2016. Anthropometric measurements were conducted on 1826 children. Recumbent length and standing height were measured using standard methods [25–27]. Data on WaSH behaviors and infrastructure characteristics were collected through self-report using structured survey questions (for behaviors) and direct observation (for infrastructure characteristics). Additional details on data collection procedures are available elsewhere [10,24,25].

We opted to use a dataset containing complete information for all dependent and independent variables, avoiding the use of imputation techniques. A total of 630 children with available anthropometric data were excluded from this analysis due to missing data on one or more independent variables. As a result, our analytic dataset consisted of 1196 children with no missing data on independent variable features, representing 65.5% of children with anthropometric measurements.

A flowchart showing the sample selection process is provided in S1 Fig of the Supporting Information section. Of the 1,826 children with anthropometric measurements in the original study, 630 (34.5%) were excluded due to missing data on one or more independent variables, resulting in an analytic dataset of 1,196 children with complete information on all WaSH and demographic covariates. This complete-case analysis approach was selected to optimize machine learning model performance, as tree-based algorithms like XGBoost and classification trees perform best with observed rather than imputed data patterns, ensuring reliable prediction accuracy for practical applications in programmatic settings. We acknowledge that this approach may introduce selection bias if data are not missing completely at random. The pattern and mechanism of missingness in our dataset may systematically differ from complete observations, potentially limiting generalizability. This complete-case analysis approach is most applicable to programmatic settings where systematic data collection on key WaSH and demographic variables is feasible.

## Outcome variable

The primary outcome was the machine learning models' ability to accurately predict stunting in children. Stunting was calculated based on length/height-for-age z score (LAZ/HAZ). According to the 2006 World Health Organization (WHO) criteria, a child is considered stunted if their LAZ/HAZ score is more than 2 standard deviations below the WHO growth standard [1]. In our analysis, stunting is represented as a binary variable: children identified as stunted are assigned a code of 1, while others are coded as 0.

## Water, sanitation, and hygiene covariates

The WaSH covariates used for algorithm development are presented in Table 1, which also classifies whether each variable is binary, categorical, or continuous. We chose these covariates to align with the primary objective of our study and to maintain consistency with the covariates utilized in the original primary research initiative [10,24].

Details of the WaSH variables have been previously described [10,24]. Briefly, standard definitions from the WHO/UNICEF Joint Monitoring Programme (JMP) for Water Supply, Sanitation and Hygiene were used to create variables for improved sanitation coverage and presence of a handwashing station. Usual defecation location was categorized for each household member; for children under 5 years old, disposal location of child feces was used in lieu of defecation location. The proportion of household members using an improved toilet for defecation (for those over 5 years old) or for child feces disposal (for those under 5 years old) was calculated to determine household sanitation use. Piped water coverage was

**Table 1. Water, sanitation, and hygiene covariates used for algorithm development.**

| Mediator | Covariate | Classification |
|---|---|---|
| Water | On-premise piped water | Binary variable |
| | | 0 = No |
| | | 1 = Yes |
| | Water source availability | Binary variable |
| | | 0 = No reported interruption in service |
| | | 1 = Any reported interruption- 24 hrs/2 weeks or anytime in previous 24hrs |
| | Drinking water storage | Categorical variable |
| | | 1 = No storage |
| | | 2 = 'Safe' storage in narrow-mouthed covered container |
| | | 3 = 'Unsafe' storage in wide-mouthed or uncovered container |
| Sanitation | Improved sanitation | Binary variable |
| | | 0 = Unimproved sanitation |
| | | 1 = Improved sanitation |
| | Proportion of household members using improved sanitation | Continuous variable with values between 0 and 1 |
| Hygiene | Handwashing station | Binary variable |
| | | 0 = No hand washing station |
| | | 1 = Hand washing station with water and soap/ash |

defined as having a piped water source within the household premises. Availability of the preferred drinking water source was a binary variable created from two questions that asked whether the household had experienced source unavailability for at least one full day in the previous 2 weeks, or for any amount of time in the previous 24 hours. Drinking water storage practices were categorized as no storage, safe storage in a covered narrow-mouthed container (with a diameter less than 6 cm), or unsafe storage.

## Demographic covariates

The demographic covariates used for algorithm development are presented in Table 2, which also classifies whether each variable is binary or categorical. We chose these covariates by examining their correlation with the intervention and anthropometric measurements, as well as based on previous analyses and through a review of the literature.

The covariates in our analysis were: (i) female caregiver education (classified as primary or less, compared to more than primary schooling), (ii) household caste/tribe (classified as scheduled caste, scheduled tribe, other backward caste, or other caste), (iii) ownership of any livestock (including poultry, small, or large livestock), (iv) optimal child feeding (classified as optimal if the caregiver reported that a child aged 6–59 months consumed at least four food groups over the past 24 hours, or that a child aged less than 6 months was exclusively breast-fed; non-optimal feeding represents caregiver-reported consumption of less than four food groups over the past 24 hours by a child 6–59 month old, or non-exclusive breastfeeding for an infant less than 6 months old), (v) standardized household wealth index, calculated using principal components analysis (PCA) as described previously [24]. In this previous study, PCA was performed on variables including household asset ownership (chair, table, refrigerator, mattress, pressure cooker, scooter or motorcycle,

**Table 2. Covariates for algorithm development.**

| Confounder | Classification |
|---|---|
| Female caregiver education | Binary variable |
| | 0 = Less than or equal to 5 years of schooling |
| | 1 = More than 5 years of schooling |
| Household caste/tribe | Categorical variable |
| | 1 = Scheduled caste |
| | 2 = Scheduled tribe |
| | 3 = Other backward caste |
| | 4 = Other caste |
| Livestock ownership | Binary variable |
| | 0 = No |
| | 1 = Yes |
| Optimal child feeding | Binary variable |
| | 0 = No |
| | 1 = Yes |
| Household wealth index | Categorical variable |
| | 1 = Poorest |
| | 2 = Poor |
| | 3 = Middle |
| | 4 = Rich |
| | 5 = Richest |
| Village status | Binary variable |
| | 0 = Control |
| | 1 = Intervention |
| Child gender | Binary variable |
| | 1 = Male |
| | 2 = Female |
| Child age | Categorical variable |
| | 1 = 0–6 mo |
| | 2 = >6–12 mo |
| | 3 = >12–24 mo |
| | 4 = >24–60 mo |

mobile phone, electric fan, sewing machine and television), housing characteristics, agricultural land acreage owned and below poverty-line status [24], (vi) village status (classified as intervention or control), (vii) child's sex, (viii) child's age.

## Machine learning analysis

**Tools.** Python (version 3.12) libraries including pandas, scikit-learn, and imblearn were utilized for data preprocessing and to build the machine learning models.

**Preprocessing.** The categorical variables were transformed into binary vectors using the one-hot encoding technique, and standard scaling was applied to the continuous variable. To address the class imbalance in our dataset (34.6% stunted vs. 65.4% non-stunted children), we employed K-fold cross-validation [28] and the Synthetic Minority Over-sampling Technique (SMOTE) [29].

K-fold cross-validation is a technique that divides the dataset into k equal-sized folds, where each fold is used as a testing set while the remaining folds are used for training. This process is repeated k times, with each fold serving as the testing set exactly once. By using k-fold cross-validation, we can obtain a more reliable estimate of the model's performance and reduce the risk of overfitting. In this study, we utilized 10-fold cross-validation to assess the performance of the machine learning algorithms.

SMOTE works by creating synthetic examples of the minority class (stunted children) in feature space rather than simply duplicating existing examples. The algorithm selects a minority class instance and its *k* nearest neighbors, then generates new synthetic instances along the line segments joining the selected instance to its neighbors. This approach provides the learning algorithm with a more balanced distribution of classes, potentially improving the model's ability to correctly classify minority instances. We applied SMOTE only to the training data in each fold of cross-validation, ensuring that the test data remained unmodified to provide an unbiased evaluation of model performance.

To address potential correlations between covariates, we examined the variance inflation factor (VIF) for all features [30]. The VIF analysis showed that no variables exceeded a VIF of 5 (highest values: improved sanitation = 4.25, proportion of household members using improved sanitation = 3.81), suggesting that multicollinearity was not a major concern in our dataset. While alternative approaches such as orthogonalization techniques exist [31], we maintained the original features to preserve interpretability, which was crucial for deriving actionable public health insights. Additionally, as subsequently described, the regularization techniques applied in our models (L2 penalty in logistic regression and built-in regularization in XGBoost) help mitigate the effects of correlated predictors.

**Feature engineering.** To identify the most relevant features for predicting the outcome variable while considering the potential presence of irrelevant or redundant features, four feature selection techniques were employed. These techniques included Structural Equation Modeling (SEM) [32], forward selection [33], backward elimination [34], and Least Absolute Shrinkage and Selection Operator (LASSO) [35]. We compared both theory-driven (SEM) and data-driven (forward selection, backward elimination, LASSO) approaches to feature selection for two reasons: (1) to assess whether theoretical knowledge from causal modeling (SEM) improves predictive performance compared to purely algorithmic selection, and (2) to determine which combination of feature engineering and machine learning algorithm yields optimal predictive accuracy for practical application. This comparison allows us to evaluate the trade-off between interpretability (theory-driven) and predictive performance (data-driven). Each feature selection technique provided its unique recommendations for the features to be dropped in predictive models.

SEM is a multivariate statistical framework that combines factor analysis and multiple regression to analyze structural relationships between measured variables and latent constructs [32]. SEM allows for the simultaneous estimation of multiple and interrelated dependencies, accounting for measurement error and enabling the modeling of complex causal pathways [36]. In our previous work [24], SEM was used to untangle direct and indirect pathways between WaSH interventions and height-for-age, and these results were utilized to inform our feature selection process by providing a set of recommended features.

Forward selection iteratively adds features to an empty model using a random forest classifier, resulting in its own set of suggested features. Backward elimination begins with a full model and iteratively removes the least significant features, yielding another set of recommended features. LASSO, a regularization technique with L1 penalty, was also used for feature selection and coefficient estimation in linear regression models, generating its specific set of selected features. The performance of the models during feature selection was evaluated using 10-fold cross-validation with AUROC as the scoring metric.

**Machine learning algorithms.** The machine learning algorithms used in our analysis include logistic regression, classification trees, support vector machines (SVM), neural networks, and extreme gradient boosting (XGBoost). These algorithms were chosen due to their proven effectiveness in binary classification tasks and their ability to handle complex relationships between the independent variables and the outcome variable. Each algorithm is summarized in Table 3, with its key characteristics, advantages, and implementation details in the context of our study.

**Table 3. Summary of machine learning algorithms.**

| Algorithm | Key Features | Advantages | Implementation Details |
|---|---|---|---|
| Logistic Regression [37] | Uses logistic function, L2 regularization (Ridge), interpretable coefficients | Simple, effective for binary classification, interpretable coefficients | Regularized with L2 penalty, trained using liblinear solver. Default regularization strength parameter C = 1.0 was used, which corresponds to $\lambda = 1.0$, as $C = 1/\lambda$. |
| Classification Tree [38] | Non-parametric, tree-like model, partitions feature space, handles categorical and numerical data | Simple to interpret, minimal preprocessing | Greedy approach to minimize impurity |
| SVM [39] | Constructs hyperplanes, maximizes margin, uses RBF kernel for non-linear decision boundaries | Effective for non-linear data, good generalization performance | RBF kernel for higher-dimensional space mapping |
| Neural Network [40] | Feedforward, input layer, two hidden layers (100 and 50 neurons), ReLU activation, Adam optimizer | Can model complex relationships, adaptive learning | Two hidden layers with ReLU, trained using Adam optimizer, max 500 iterations |
| XGBoost [41] | Combines decision trees, optimized gradient boosting, handles missing data, hyperparameter tuning | High performance, scalability, handles missing data | Hyperparameters tuned via random search with 50 iterations, stratified 10-fold cross-validation, optimizing for F1-score |

## Model validation approach

We employed 10-fold cross-validation as our primary validation strategy, which provides reliable performance estimates for model comparison within our dataset. We acknowledge that this internal validation approach, while following established best practices for algorithm selection, may result in modest overestimation of model performance compared to external validation on independent datasets. Our results therefore represent the comparative performance of different machine learning approaches under identical validation conditions, rather than definitive estimates of real-world performance. External validation in different geographic contexts and populations with varying WaSH conditions and stunting prevalence would be needed to confirm generalizability and establish definitive performance benchmarks for clinical or programmatic deployment.

**Performance measures.** Performance measures used to evaluate machine learning models in this analysis, including area under the receiver operating characteristic curve (AUROC), recall, specificity, accuracy, precision, and F1 score, are summarized in Table 4.

## Results

### Descriptive results

Table 5 presents descriptive characteristics of the analytic dataset, except for the proportion of household members using improved sanitation, which had a mean of 35%. The data showed that 35% of children were stunted. A large proportion of the children belonged to households that lacked on-premise piped water (64%), experienced interrupted water supply (85%), and used unsafe drinking water storage (77%). At the same time, 73% had a hand washing station with soap/ash and water available. Around 53% of children resided in households with improved sanitation facilities. Most children (89%) were aged between 6 and 59 months, had caregivers with at least primary education (62%), and experienced optimal feeding (59%). Children belonging to scheduled castes and tribes comprised 33% of the sample, while those from households with poor and poorest wealth indices constituted 39%.

### Feature engineering results

Table 6 presents the recommendations of four feature selection techniques used in this analysis: SEM, forward selection, backward elimination, and LASSO. While LASSO did not suggest eliminating any features, the other three techniques had

**Table 4. Summary of performance measures used to evaluate prediction models.**

| Performance Measure | Description | Formula |
|---|---|---|
| AUROC | Primary measure of a machine learning model's performance. The receiver operating characteristic curve plots sensitivity against the false positive rate (= 1 − specificity) at different classification thresholds. The area under this curve provides an aggregate measure of the classifier's performance across all possible classification thresholds | *NA* |
| Recall | Also known as sensitivity, denotes the percentage of actual positive class instances that are correctly detected by the model | $Recall = \dfrac{True\ Positives}{True\ Positives\ +\ False\ Negatives}$ |
| Specificity | Measures the proportion of actual negative class instances that are correctly predicted as negative by the model | $Specificity = \dfrac{True\ Negatives}{True\ Negatives\ +\ False\ Positives}$ |
| Accuracy | Represents the overall correctness of the model's predictions, calculated as the ratio of correct predictions to the total number of instances evaluated | $Accuracy = \dfrac{True\ Positives\ +\ True\ Negatives}{True\ Positives\ +\ False\ Negatives\ +\ True\ Negatives\ +\ False\ Positives}$ |
| Precision | Proportion of positive class predictions that belong to the positive class. It measures the likelihood that a randomly selected instance predicted as positive truly belongs to the positive class | $Precision = \dfrac{True\ Positives}{True\ Positives\ +\ False\ Positives}$ |
| F1 Score | Harmonic mean of precision and recall. It combines both metrics to provide a balanced measure, as precision and recall can share an inverse relationship | $F1\ Score = \dfrac{2\ \times\ Precision\ \times\ Recall}{Precision\ +\ Recall}$ |

varying recommendations. The "drinking water storage" feature was recommended for removal by SEM, forward selection, and backward elimination. Additionally, both forward selection and backward elimination recommended the exclusion of features related to "Proportion of household members using improved sanitation" and "village status." SEM and forward selection techniques suggested removing the feature representing "optimal child feeding".

## Model performance results

To assess the performance of the various prediction models, we evaluated them across several key metrics including AUROC, recall, specificity, accuracy, precision and F1 score. The models were built using different feature engineering recommendation (per SEM, forward selection, backward elimination, and LASSO (no feature removal)) and machine learning algorithms (logistic regression, classification trees, SVM, neural networks, and XGBoost). Table 7 summarizes the performance of the machine learning models.

While XGBoost demonstrated superior performance, we acknowledge the potential for overfitting, particularly in complex models. We addressed this concern through several approaches: (1) 10-fold cross-validation to provide robust performance estimates, (2) hyperparameter tuning via random search with stratified cross-validation, and (3) XGBoost's built-in regularization parameters. However, as discussed in our Methods and Limitations sections, external validation on independent datasets would be needed to fully assess generalizability and confirm that performance estimates are not inflated.

Fig 1 illustrates the comparative performance of all five algorithms across the four feature engineering approaches. Across all the approaches, the XGBoost model consistently achieved the best performance. The XGBoost model using features selected via forward selection achieved an AUROC of 0.96, recall of 0.81, specificity of 0.92, accuracy of 0.88, precision of 0.85 and F1 score of 0.83. This model far surpassed the other algorithms. Similarly, the XGBoost models using backward elimination (AUROC 0.89) or all features (AUROC 0.85) outperformed the other model types.

**Table 5.  Descriptive characteristics of dataset.**

| Characteristic | Stunted | % | Not Stunted | % | All | % |
|---|---|---|---|---|---|---|
| Stunted | | | | | | |
| No | 0 | 0 | 782 | 100 | 782 | 65.4 |
| Yes | 414 | 100 | 0 | 0 | 414 | 34.6 |
| On-premise piped water | | | | | | |
| No | 296 | 71.5 | 467 | 59.7 | 763 | 63.8 |
| Yes | 118 | 28.5 | 315 | 40.3 | 433 | 36.2 |
| Water source availability | | | | | | |
| No reported interruption in service | 52 | 12.6 | 126 | 16.1 | 178 | 14.9 |
| Any reported interruption- 24 hrs/2 weeks or anytime in previous 24hrs | 362 | 87.4 | 656 | 83.9 | 1018 | 85.1 |
| Drinking water storage | | | | | | |
| No storage | 9 | 2.2 | 15 | 1.9 | 24 | 2 |
| 'Safe' storage in narrow-mouthed covered container | 77 | 18.6 | 169 | 21.6 | 246 | 20.6 |
| 'Unsafe' storage in wide-mouthed or uncovered container | 328 | 79.2 | 598 | 76.5 | 926 | 77.4 |
| Improved sanitation | | | | | | |
| Unimproved sanitation | 248 | 59.9 | 316 | 40.4 | 564 | 47.2 |
| Improved sanitation | 166 | 40.1 | 466 | 59.6 | 632 | 52.8 |
| Handwashing station | | | | | | |
| No hand washing station | 160 | 38.6 | 169 | 21.7 | 329 | 27.5 |
| Hand washing station with water and soap/ash | 254 | 61.4 | 613 | 78.4 | 867 | 72.5 |
| Female caregiver education | | | | | | |
| Less than or equal to 5 years of schooling | 214 | 51.7 | 247 | 31.6 | 461 | 38.5 |
| More than 5 years of schooling | 200 | 48.3 | 535 | 68.4 | 735 | 61.5 |
| Household caste/tribe | | | | | | |
| Scheduled caste | 110 | 26.6 | 104 | 13.3 | 214 | 17.9 |
| Scheduled tribe | 95 | 22.9 | 83 | 10.6 | 178 | 14.9 |
| Other backward caste | 133 | 32.1 | 322 | 41.2 | 455 | 38 |
| Other caste | 76 | 18.4 | 273 | 34.9 | 349 | 29.2 |
| Livestock ownership | | | | | | |
| No | 240 | 58 | 459 | 58.7 | 699 | 58.4 |
| Yes | 174 | 42 | 323 | 41.3 | 497 | 41.6 |
| Optimal child feeding | | | | | | |
| No | 193 | 46.6 | 294 | 37.6 | 487 | 40.7 |
| Yes | 221 | 53.4 | 488 | 62.4 | 709 | 59.3 |
| Household wealth index | | | | | | |
| Poorest | 129 | 31.2 | 120 | 15.3 | 249 | 20.8 |
| Poor | 96 | 23.2 | 124 | 15.9 | 220 | 18.4 |
| Middle | 89 | 21.5 | 150 | 19.2 | 239 | 20 |
| Rich | 68 | 16.4 | 178 | 22.8 | 246 | 20.6 |
| Richest | 32 | 7.7 | 210 | 26.9 | 242 | 20.2 |
| Village status | | | | | | |
| Control | 240 | 58 | 391 | 50 | 631 | 52.8 |
| Intervention | 174 | 42 | 391 | 50 | 565 | 47.2 |
| Child gender | | | | | | |
| Male | 213 | 51.4 | 391 | 50 | 604 | 50.5 |
| Female | 201 | 48.6 | 391 | 50 | 592 | 49.5 |

*(Continued)*

**Table 5.** (Continued)

| Characteristic | *Stunted* | *%* | *Not Stunted* | *%* | *All* | *%* |
|---|---|---|---|---|---|---|
| Child age | | | | | | |
| 0–6 mo | 17 | 4.1 | 30 | 3.8 | 47 | 3.9 |
| >6–12 mo | 24 | 5.8 | 56 | 7.2 | 80 | 6.7 |
| >12–24 mo | 97 | 23.4 | 184 | 23.5 | 281 | 23.5 |
| >24–60 mo | 276 | 66.7 | 512 | 65.5 | 788 | 65.9 |

**Table 6. Recommendations of feature engineering techniques.**

| Feature engineering technique | Recommendation to remove features |
|---|---|
| SEM [24] | Water source availability, Drinking water storage, Handwashing station, Optimal child feeding, Livestock ownership |
| Forward Selection | Drinking water storage, Proportion of household members using improved sanitation, Village status, Optimal child feeding |
| Backward Elimination | Water source availability, Village status, Proportion of household members using improved sanitation, Drinking water storage |
| LASSO | None |

**Table 7. Evaluation of algorithms trained to predict stunting.**

| Feature Eng. | Prediction Algorithm | Performance Measure | | | | | |
|---|---|---|---|---|---|---|---|
| | | AUROC (95% CI) | Recall | Specificity | Accuracy | Precision | F1 |
| **SEM** | Logistic Regression | 0.688 (0.590–0.784) | 0.642 | 0.611 | 0.622 | 0.467 | 0.539 |
| | Classification Tree | 0.588 (0.485–0.688) | 0.400 | 0.723 | 0.612 | 0.436 | 0.416 |
| | SVM | 0.665 (0.564–0.766) | 0.591 | 0.640 | 0.623 | 0.467 | 0.520 |
| | Neural Network | 0.613 (0.508–0.719) | 0.480 | 0.654 | 0.594 | 0.423 | 0.447 |
| | XGBoost | 0.789 (0.763–0.813) | 0.584 | 0.809 | 0.731 | 0.618 | 0.601 |
| **Forward Sel.** | Logistic Regression | 0.681 (0.581–0.777) | 0.640 | 0.613 | 0.622 | 0.469 | 0.538 |
| | Classification Tree | 0.586 (0.486–0.686) | 0.422 | 0.739 | 0.629 | 0.465 | 0.441 |
| | SVM | 0.665 (0.562–0.766) | 0.582 | 0.662 | 0.634 | 0.478 | 0.523 |
| | Neural Network | 0.643 (0.537–0.745) | 0.497 | 0.704 | 0.632 | 0.472 | 0.482 |
| | XGBoost | 0.959 (0.949–0.968) | 0.811 | 0.928 | 0.888 | 0.857 | 0.833 |
| **Backward Elim.** | Logistic Regression | 0.687 (0.588–0.782) | 0.642 | 0.616 | 0.625 | 0.471 | 0.542 |
| | Classification Tree | 0.573 (0.474–0.675) | 0.381 | 0.758 | 0.627 | 0.457 | 0.415 |
| | SVM | 0.670 (0.570–0.772) | 0.563 | 0.686 | 0.643 | 0.488 | 0.522 |
| | Neural Network | 0.633 (0.527–0.738) | 0.470 | 0.719 | 0.633 | 0.474 | 0.471 |
| | XGBoost | 0.889 (0.870–0.908) | 0.678 | 0.904 | 0.826 | 0.789 | 0.729 |
| **None** | Logistic Regression | 0.678 (0.578–0.775) | 0.637 | 0.606 | 0.616 | 0.462 | 0.534 |
| | Classification Tree | 0.563 (0.469–0.657) | 0.402 | 0.713 | 0.606 | 0.426 | 0.413 |
| | SVM | 0.664 (0.562–0.766) | 0.545 | 0.681 | 0.634 | 0.476 | 0.506 |
| | Neural Network | 0.630 (0.528–0.734) | 0.446 | 0.709 | 0.618 | 0.453 | 0.447 |
| | XGBoost | 0.848 (0.824–0.869) | 0.654 | 0.849 | 0.781 | 0.696 | 0.675 |

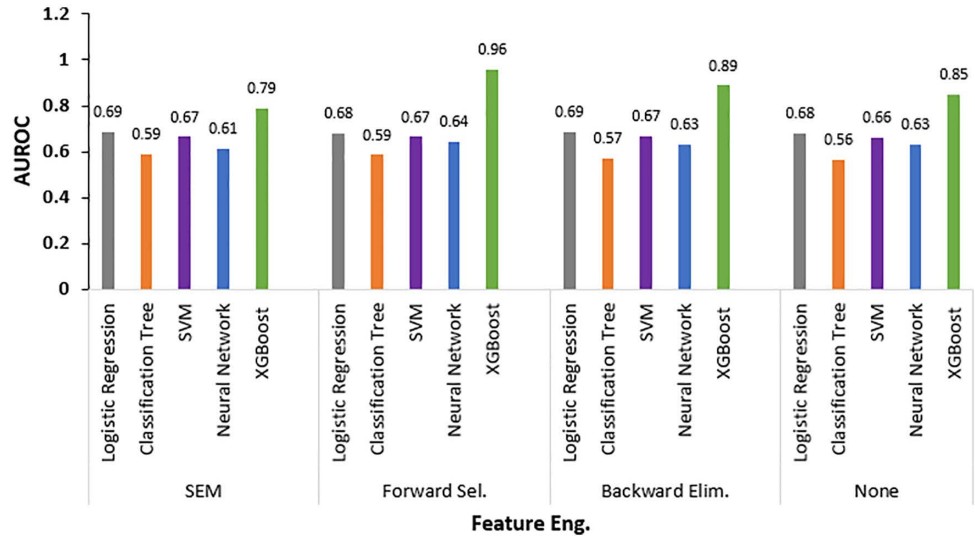

**Fig 1. Comparative AUROCs of algorithms across feature engineering.**

The impact of using no feature engineering versus various feature selection techniques can be observed in Table 7. When no feature engineering was performed (labeled as "None" in Table 7), the XGBoost model achieved an AUROC of 0.848, which was substantially lower than its performance with forward selection (AUROC 0.959). This improvement demonstrates the significant value of appropriate feature selection for complex algorithms like XGBoost. In contrast, simpler models showed minimal sensitivity to feature engineering. For instance, logistic regression maintained nearly identical performance with no feature engineering (AUROC 0.678) compared to any feature selection technique (AUROC range: 0.681–0.688). Classification trees performed slightly better with no feature engineering (AUROC 0.563) than with SEM-based feature selection (AUROC 0.558). SVM and neural networks showed modest improvements with feature selection compared to using all features, but these gains were much less dramatic than those observed with XGBoost. These comparative results highlight that while feature engineering is critical for optimizing complex non-linear models like XGBoost, it provides limited benefits for simpler, more rigid algorithms in this application.

Among the other algorithms, logistic regression generally performed the next best, followed by SVM, neural networks, and classification trees. For logistic regression, the choice of feature selection approach did not make a large difference, with AUROCs ranging from 0.68–0.69. Feature selection led to a slight improvement in the performance of SVMs. Specifically, using forward selection or backward elimination to choose features resulted in AUROCs between 0.66 and 0.67, which was marginally better than the AUROC of 0.66 achieved when using all available features. Neural networks saw a slight improvement with feature selection (AUROCs 0.63–0.64) compared to using all features (AUROC 0.63). Classification trees performed the poorest overall and did not benefit from feature selection. Overall, the choice of feature selection had less impact overall compared to the choice of algorithm.

## WaSH-specific results

The XGBoost model with forward selection retained four WaSH variables that were most predictive of stunting in this sample: improved sanitation coverage, presence of a handwashing station, piped water coverage, and availability of preferred drinking water source. The model also retained six demographic covariates (female caregiver education, household caste/tribe, livestock ownership, household wealth, child's gender, and child's age). The forward selection feature selection led

to dropping two WaSH variables (drinking water storage and proportion of household members using improved sanitation) and two demographic covariates (village status and optimal child feeding), which had lower predictive power for stunting in this sample.

## Discussion

This study demonstrates that machine learning algorithms can accurately predict child stunting based on WaSH behaviors and infrastructure, with four WaSH variables having the strongest predictive power: improved sanitation coverage, presence of a handwashing station, piped water coverage, and availability of preferred drinking water source. Among the combinations tested, XGBoost consistently outperformed the other algorithms, regardless of the feature engineering technique used. The highest performance was achieved by XGBoost with forward selection, suggesting that forward selection, which iteratively adds the most informative features to the model, successfully identified the most relevant predictors for stunting. XGBoost's superior performance across all feature engineering approaches can be attributed to several factors. First, XGBoost builds sequential decision trees that learn from the errors of previous trees (gradient boosting), making it particularly effective at capturing non-linear relationships and interactions between WaSH variables. Second, XGBoost has built-in regularization parameters that help prevent overfitting, which is especially important in our relatively small dataset. Third, XGBoost handles mixed data types efficiently, accommodating our combination of binary, categorical, and continuous WaSH variables. The XGBoost model's exceptional accuracy in predicting stunting is consistent with prior research indicating XGBoost's effectiveness in stunting prediction [20–22].

While XGBoost demonstrated superior predictive performance, the choice of algorithm for practical implementation involves important trade-offs between accuracy and interpretability. XGBoost's "black box" nature means that individual predictions are difficult to explain to program staff or beneficiaries, which may limit trust and uptake in field settings. Simpler models like logistic regression, despite lower accuracy, provide transparent coefficient estimates that clearly show how each WaSH factor influences risk. For large-scale programmatic applications where stakeholder buy-in and explainability are critical, the interpretability-accuracy trade-off warrants careful consideration. In settings prioritizing maximum predictive accuracy for resource-constrained targeting decisions, XGBoost's superior performance may justify its complexity. However, in settings where program staff need to understand and communicate why certain children are classified as high-risk, simpler interpretable models may be preferable despite lower accuracy. Future implementations should consider hybrid approaches, such as using XGBoost for initial risk scoring while developing simplified decision rules for field communication.

Regarding forward selection's effectiveness, this technique likely performed well because it prioritizes features with the strongest predictive power while systematically excluding redundant or uninformative variables. Forward selection's iterative approach aligns well with the gradient boosting process of XGBoost, potentially creating synergy between the feature selection method and the algorithm. Furthermore, forward selection identified a parsimonious set of features that reduced noise in the data, allowing the XGBoost algorithm to focus on the most informative signals for stunting prediction.

The other algorithms, including logistic regression, classification tree, SVM, and neural network, showed moderate performance across all feature engineering techniques. Their AUROCs ranged from 0.563 to 0.688, indicating that they were less effective in predicting stunting compared to XGBoost.

One of the approaches adopted in our study, employing SEM for feature engineering, builds upon an earlier study [24] where SEM was utilized to examine the interconnected pathways within combined WaSH interventions for the same study population as in this analysis. Our previous SEM analysis [24] identified significant pathways from improved sanitation coverage, piped water coverage, and drinking water availability to height-for-age z-scores, mediated through household sanitation use. The SEM approach revealed that increased use of improved sanitation facilities was the most proximal determinant of improved HAZ, with village-level sanitation coverage having both direct effects and indirect effects (through household use) on child growth outcomes.

In the current study, we used these SEM insights to inform feature selection, leveraging the identified mechanistic pathways. The interpretative value of the SEM approach lies in its ability to distinguish between direct and indirect effects and to model the sequential nature of WaSH pathways affecting child growth. For example, the SEM results suggested that handwashing stations may influence child growth through indirect pathways related to reduced pathogen transmission, rather than having a strong direct effect. These insights complement the predictive power of our machine learning models by providing potential explanations for why certain WaSH factors emerge as important predictors.

Integrating SEM into our feature engineering process offers a valuable avenue for exploring intricate WaSH behaviors and infrastructure impacts, thus facilitating well-informed decisions regarding feature selection or removal. Notably, in this analysis, the combination of SEM and XGBoost, despite yielding a lower AUROC (0.78) compared to other XGBoost combinations, still demonstrates robust predictive performance. The strength of this combination lies in its potential to deliver a more interpretable model by leveraging SEM insights to guide feature selection and elucidate the relationships between WaSH factors and stunting. Through the incorporation of domain expertise and theory-driven methodologies such as SEM into the machine learning pipeline, researchers can enhance the interpretability and credibility of predictive models, thereby facilitating their adoption and applicability in real-world scenarios [24].

The fact that the present study uses the same original dataset as a previous SEM analysis [24] offers a novel opportunity to compare results produced using different methodologies. It is worth noting that the previous study used HAZ as the outcome variable rather than stunting. Still, comparing the results can generate insights into the utility of the machine learning approach. Such comparisons can also highlight the complementary strengths of statistical and machine learning approaches in understanding and predicting a complex outcome such as stunting. In the previous study, the SEM analysis indicated that village intervention status, improved sanitation coverage, piped water coverage, and drinking water availability all had significant indirect effects on HAZ through household sanitation use. In the current study, all four of these variables were retained in the XGBoost with forward selection model, except for village intervention status. In both studies, drinking water storage was determined to be unrelated to HAZ or stunting. Both studies also included female caregiver education, household caste/tribe, livestock ownership, and household wealth as covariates, but in the current study, the forward selection procedure suggested dropping the child feeding variable, which had been included as a covariate in the previous study. Finally, in the current study, the final XGBoost with forward selection model retained presence of a handwashing station, which was not associated with HAZ in the earlier study.

The concordant results from the current study and the previous SEM study reinforce the importance of three WaSH variables for child nutritional status: improved sanitation coverage, piped water coverage, and drinking water availability. These variables have also been consistently identified in the literature, particularly in Odisha, where multiple studies have found improved sanitation to be critical for reductions in stunting [10,42,43]. It is important to note that large randomized controlled trials in other settings have observed no effect of improvements in sanitation coverage on child stunting [13–15]. Researchers have hypothesized that this lack of effect on child growth may be because the WaSH interventions provided in those trials consisted of basic, low-cost infrastructure such as pit latrines, which may be insufficiently effective for reducing fecal contamination in the environment [44,45]. Partly in response to those trials, recommendations now focus on comprehensive, 'transformative' WaSH approaches, including high community-level coverage of improved sanitation combined with continuous and convenient access to drinking water [45]. Our results provide additional support for the importance of combined interventions that increase coverage of improved sanitation and piped drinking water that is consistently available. Still, our results are focused on predictive modeling, which may differ from intervention effects under real-world conditions; gold standard evidence from randomized controlled trials of 'transformative' WaSH approaches would further strengthen the evidence base related to causal linkages between WaSH and child stunting.

It is critical to emphasize that the strong predictive associations identified in our machine learning models do not necessarily imply that interventions targeting these factors will produce equivalent improvements in child growth. Our models identify children at high risk based on observed patterns in cross-sectional data, which reflects the complex interplay of

WaSH conditions with numerous other factors (socioeconomic status, dietary patterns, maternal healthetc.) within our study population. In contrast, intervention trials isolate the causal effect of specific WaSH improvements. The mixed evidence from large randomized controlled trials [13–15], which found no effect of basic WaSH interventions on stunting despite observational associations, illustrates this prediction-causation gap. Our predictive models are valuable for identifying which children currently face highest risk and thus may benefit from comprehensive support, but they cannot predict the magnitude of benefit from any specific intervention. This distinction is fundamental: high predictive accuracy indicates we can identify vulnerable children, not that we know how to effectively intervene.

Our results also reinforce the importance of several demographic variables for child growth, which were retained in the XGBoost model: female caregiver education, household caste/tribe, livestock ownership, household wealth, child's gender, and child's age. Globally, extensive evidence exists for the role of female caregiver education and household wealth as basic or enabling determinants of child nutritional status [46–48]. Similarly, household caste/tribe status and livestock ownership may be closely linked to household wealth [49,50]. The child's gender may be important due to gendered social norms related to child feeding, particularly in India [51,52]. Finally, child age is known to predict linear growth, as most linear growth faltering occurs in the first 1000 days of life [48]. While these results align well with existing evidence, it was surprising that the XGBoost model recommended dropping the variable for optimal child feeding, given the clear importance of child feeding practices for growth. This exemplifies a key distinction between predictive modeling and causal inference frameworks. In our predictive approach, XGBoost prioritized variables that maximize predictive accuracy rather than establishing causal importance. The complex non-linear relationships and interactions captured by XGBoost may have determined that other variables in the model collectively provided stronger predictive signals for stunting outcomes in this particular dataset, even while child feeding remains theoretically important.

Our study has several implications for future research and practice. In particular, the high predictive accuracy of the XGBoost model demonstrates its potential as a valuable tool for public health practitioners. By inputting readily available WaSH data into this model, practitioners could identify high-risk children or households for prioritized interventions. This approach could enhance the efficiency and impact of stunting prevention programs, particularly in resource-limited settings where targeted interventions are crucial.

This study's limitations include the cross-sectional nature of the data, which precludes causal inferences, and the specific geographical context, which may limit the generalizability of the findings. The geographic specificity of our study (rural Odisha, India) limits the direct extrapolation of our findings to other contexts. While the machine learning methodology demonstrated here is broadly applicable, the specific predictive relationships between WaSH factors and stunting may vary across contexts. The determinants of stunting are complex, multi-level, and interconnected [53] and, therefore, may exhibit different predictor importance rankings by location. For example, access to improved sanitation varies widely between and within LMICs [54]. Hence, the importance of improved sanitation coverage as a predictor of stunting is likely to vary as well. Future research should validate these machine learning approaches using locally relevant datasets to ensure culturally and contextually appropriate risk prediction models. Furthermore, the machine learning models do not currently incorporate different grades or severity levels of stunting, which could be addressed in future iterations to provide more nuanced risk predictions. The moderate performance of algorithms other than XGBoost, indicates that there is still room for improvement in terms of model selection and optimization.

While XGBoost demonstrated superior predictive performance in our study, it has limitations. Despite our use of k-fold cross-validation and hyperparameter tuning, XGBoost models remain susceptible to overfitting. The model's performance is also dependent on the specific characteristics of our dataset, and these results may not generalize to populations with different demographic profiles or WaSH conditions. Future research could include validating the model using different datasets from varying geographical contexts to further assess generalizability and potential overfitting issues. Our cross-validation approach, while following established best practices, may result in modest overestimation of model performance compared to external validation. However, this potential limitation does not undermine our primary contribution

of demonstrating the comparative effectiveness of different machine learning algorithms under identical validation conditions. Additionally, exploring the use of ensemble methods, which combine multiple algorithms, could improve predictive performance.

Another limitation is our use of the complete-case analysis approach. The exclusion of 35% of children due to missing data raises important considerations about potential selection bias. If missing data patterns are systematic, implications may follow. First, our model's performance estimates may be optimistic if trained on a non-representative subsample. Second, the model may perform differently when applied to populations with characteristics different from our analytical sample. Third, the predictive importance of specific WaSH variables may differ across populations with different baseline conditions. For example, a WaSH factor may be highly predictive in populations with moderate access (where meaningful variation exists) but less predictive in populations with universally low or universally high access. These considerations underscore the need for external validation across diverse populations and performance monitoring that specifically tracks accuracy across different subgroups defined by baseline WaSH and socioeconomic conditions. The complete-case analysis approach, while methodologically appropriate for machine learning applications, means our findings are most directly applicable to populations where systematic WaSH and demographic data collection is feasible. While we cannot rule out potential selection bias from excluding observations with missing data, this represents a common scenario in well-designed program implementation contexts. Future validation studies across diverse populations and data collection contexts would further establish the broader applicability of our predictive modeling approach.

Future research is needed to validate these findings in other populations, refine the models by incorporating additional risk factors such as community water and sanitation characteristics [55] as well as pollution [56,57], and assess the impact of machine learning-informed interventions on child growth outcomes. The development of data-driven, individualized risk prediction tools could contribute to the creation of a more effective, efficient, and equitable public health approach to promoting optimal child growth and development.

## Conclusions

Machine learning algorithms, particularly the XGBoost model, show promise for predicting the risk of childhood stunting using WaSH behaviors and infrastructure in rural Odisha, India. In our dataset, the XGBoost model with forward selection achieved high predictive accuracy (AUROC 0.959), identifying four key WaSH variables: improved sanitation coverage, presence of a handwashing station, piped water coverage, and availability of preferred drinking water source. The choice between performance and interpretability should be based on the specific requirements of the application, considering the trade-off between the superior performance of XGBoost and the interpretability of models like SEM.

If validated in diverse geographic and demographic contexts, such machine learning models could be operationalized through practical screening tools for WaSH programs. For example, field staff could use simple data collection forms capturing the four key WaSH variables (along with relevant demographic factors) to generate risk scores for enrolled children. These scores could inform programmatic decisions such as: (1) prioritizing households with highest-risk children for intensive WaSH interventions when resources are limited, (2) targeting behavior change communication to families of high-risk children, and (3) allocating follow-up monitoring visits based on predicted risk levels. Such risk-based targeting could improve program efficiency by focusing resources where they are most likely to impact child growth outcomes.

However, certain limitations must be acknowledged. Our findings are based on a single geographic context and require external validation across different settings with varying WaSH conditions, stunting prevalence, and cultural contexts before broad implementation. The complete-case analysis approach excluded 35% of children with anthropometric measurements due to missing data on WaSH or demographic variables; while appropriate for algorithm development, this means our model performs best in settings with systematic data collection capabilities and may not generalize to populations with incomplete data. Most critically, these predictive models identify children at risk based on observed patterns but do not establish that interventions targeting these predictors will necessarily improve outcomes- rigorous evaluation of

intervention effectiveness remains essential. Future research should focus on external validation in diverse populations, testing of practical implementation strategies, and assessment of whether ML-informed targeting improves program outcomes and child health. The integration of machine learning techniques, particularly XGBoost, into WaSH program planning has potential to enhance the identification of high-risk children and the targeting of interventions, potentially improving the efficiency and effectiveness of efforts to reduce childhood stunting in resource-limited settings where validated models and appropriate data infrastructure exist.

## Supporting information

**S1 Fig. Sample inclusion/exclusion flow chart.**
(TIF)

## Acknowledgments

We thank the study team and participants in the original matched cohort study, without whom this work would not be possible. We also appreciate Shubhayu Sinharoy for his advice and insights during the project.

## Author contributions

**Conceptualization:** Sanaya Sinharoy, Heather Reese, Thomas Clasen, Sheela S. Sinharoy.

**Data curation:** Sanaya Sinharoy, Heather Reese, Thomas Clasen, Sheela S. Sinharoy.

**Formal analysis:** Sanaya Sinharoy, Heather Reese, Thomas Clasen, Sheela S. Sinharoy.

**Funding acquisition:** Heather Reese, Thomas Clasen, Sheela S. Sinharoy.

**Investigation:** Sanaya Sinharoy, Heather Reese, Thomas Clasen, Sheela S. Sinharoy.

**Methodology:** Sanaya Sinharoy, Heather Reese, Thomas Clasen, Sheela S. Sinharoy.

**Project administration:** Sanaya Sinharoy, Heather Reese, Thomas Clasen, Sheela S. Sinharoy.

**Resources:** Sanaya Sinharoy, Heather Reese, Thomas Clasen, Sheela S. Sinharoy.

**Software:** Sanaya Sinharoy, Heather Reese, Thomas Clasen, Sheela S. Sinharoy.

**Supervision:** Sanaya Sinharoy, Heather Reese, Thomas Clasen, Sheela S. Sinharoy.

**Validation:** Sanaya Sinharoy, Heather Reese, Thomas Clasen, Sheela S. Sinharoy.

**Visualization:** Sanaya Sinharoy, Heather Reese, Thomas Clasen, Sheela S. Sinharoy.

**Writing – original draft:** Sanaya Sinharoy, Heather Reese, Thomas Clasen, Sheela S. Sinharoy.

**Writing – review & editing:** Sanaya Sinharoy, Heather Reese, Thomas Clasen, Sheela S. Sinharoy.

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
