## [Decision Letter · Decision Letter 0]

16 Mar 2025

Dear Dr. Sinharoy,

Thank you for submitting your manuscript to PLOS ONE. After careful consideration, we feel that it has merit but does not fully meet PLOS ONE’s publication criteria as it currently stands. Therefore, we invite you to submit a revised version of the manuscript that addresses the points raised during the review process.

We look forward to receiving your revised manuscript.

Kind regards,

Ashish Wasudeo Khobragade, MD

Academic Editor

PLOS ONE

Journal Requirements:

3, Thank you for stating in your Funding Statement:

“This work was supported by the Bill & Melinda Gates Foundation [grant numbers OPP1008048 and OOP1125067].”

“This work was supported by the Bill & Melinda Gates Foundation [grant numbers OPP1008048 and OOP1125067]. We thank the study team and participants in the original matched cohort study, without whom this work would not be possible. “

“This work was supported by the Bill & Melinda Gates Foundation [grant numbers OPP1008048 and OOP1125067].”

5. In the online submission form, you indicated that [The datasets used and/or analyzed during the current study are available from the corresponding author on reasonable request.].

Additional Editor Comments:

In this study, the XGBoost model is used to predict stunting.

The authors mention moderate performance of other models in the limitations. What are the limitations of using the XGBoost model? What is the effect of overfitting and different data sets on the results?

Reviewers' comments:

Reviewer's Responses to Questions

**Comments to the Author**

1. Is the manuscript technically sound, and do the data support the conclusions?

Reviewer #1: Yes

Reviewer #2: Yes

2. Has the statistical analysis been performed appropriately and rigorously?

Reviewer #1: Yes

Reviewer #2: Yes

3. Have the authors made all data underlying the findings in their manuscript fully available?

Reviewer #1: Yes

Reviewer #2: Yes

4. Is the manuscript presented in an intelligible fashion and written in standard English?

Reviewer #1: Yes

Reviewer #2: Yes

Reviewer #1: This paper presents the application of machine learning to model the relationship between child stunting and a set of features including those related to water sanitation and hygiene in a local population in India. Appropriate stages have been followed to collect the data, select features and build and evaluate ML models. The work is overall acceptable, but with some minor changes to apply:

1- To be more informative, in Table 5, please show the stratification of the independent variables based on the stunting state.

2- Authors are recommended to add some discussion on the impact of non-WaSH demographic variables on stunting based on their machine learning modeling.

Reviewer #2: In this paper, which is a follow-up of a previous paper using the same dataset, the authors have implemented several prediction algorithms and feature engineering techniques to predict stunting in children under 5 years of age. The main aim is to identify an optimal learning algorithm. Based on several covariates, including water, sanitation, hygiene behaviors, and demographic covariates, the authors identify four variables as key factors in determining stunting—improved sanitation coverage, presence of a handwashing station, piped water coverage, and availability of a preferred drinking water source.

I have several major comments and some minor comments:

1. The advances beyond [25] are unclear. As the authors state several times, the key factors influencing stunting remain the same, and the same algorithm appears to be the most optimal. Please clarify the advances.

2. While many different algorithms were used to predict stunting and the results from each reported, there is no explanation or discussion on why they think XGBoost performed better than others or why forward selection was the optimal feature engineering technique.

3. There is also no commentary on what are the difficulties in linking WaSH interventions with improvements in child growth. Why has this link been hard to determine in the past?

4. A primary concern with linear models is that if correlations exist between the covariates, feature importance and predictions can be unstable. I wonder if the covariates used in this analysis are correlated, and if so, these should be accounted for. One can use orthogonalization techniques like QR decomposition to disassociate the variables.

5. It seems that forward selection also found some demographic covariates to be important—these should also be discussed.

6. SEM has never been explained, and references for all the algorithms used are missing.

7. Despite advocating for SEM in terms of interpretability of the features, the authors do not provide any interpretations of its results.

8. Please provide a reference for SMOTE and a brief description. It seems to be very crucial for model training.

9. How would the results change if no feature engineering was performed?

10. How was lambda determined in L2 regularization?

11. Lines 287-289: Since [25] shows similar results using ML techniques, I am not sure this claim is true.

Minor comments

1. Lines 131-132: "Availability of the preferred drinking water source was a binary variable defined as having experienced source unavailability for at least 24 hours in the previous 2 weeks, or at any time in the previous 24 hours." Are these two conditions different?

2. Line 149: "Standardized household wealth index, calculated using principal components analysis as described previously." How was this performed? A brief description would improve clarity.

3. Table 5 could be moved to the appendix, as its contribution to the main results are minor.

**Do you want your identity to be public for this peer review?** For information about this choice, including consent withdrawal, please see our Privacy Policy

Reviewer #1: **Yes:** Ebrahim Barzegari

Reviewer #2: No

---

## [Author Response · Author response to Decision Letter 1]

7 Apr 2025

Editor comment: Thank you for this important question. We have expanded our discussion of XGBoost limitations in the revised manuscript to reflect the below in the “Discussions” section:

While XGBoost demonstrated superior predictive performance in our study, it has limitations. Despite our use of k-fold cross-validation and hyperparameter tuning, XGBoost models remain susceptible to overfitting. The model's performance is also dependent on the specific characteristics of our dataset, and these results may not generalize to populations with different demographic profiles or WaSH conditions. These limitations should be considered alongside the model's strong predictive capabilities when evaluating its potential application in public health practice. Future research could include validating the model using different datasets from varying geographical contexts to further assess generalizability and potential overfitting issues.

Reviewer #1, Comment #1: Thank you for this suggestion. We have modified this table to show descriptive characteristics of the dataset stratified by stunting status (stunted vs. not stunted as two additional columns). This provides a clearer picture of how the independent variables differ between children who are stunted and those who are not.

Reviewer #1, Comment #2: We have added the following paragraph to the Discussion section, to discuss our results related to non-WaSH demographic variables:

Our results also reinforce the importance of several demographic variables for child growth, which were retained in the XGBoost model: female caregiver education, household caste/tribe, livestock ownership, household wealth, child’s gender, and child’s age. Globally, extensive evidence exists for the role of female caregiver education and household wealth as basic or enabling determinants of child nutritional status [Smith et al., 2015; Vaivada et al., 2020; Victora et al., 2021]. Similarly, household caste/tribe status and livestock ownership may be closely linked to household wealth [Zacharias et al., 2011; Khan et al., 2021]. The child’s gender may be important due to gendered social norms related to child feeding, particularly in India [Jayachandran et al., 2011; Mishra et al., 2004]. Finally, child age is known to predict linear growth, as most linear growth faltering occurs in the first 1000 days of life [Victora et al., 2021]. While these results align well with existing evidence, it was surprising that the XGBoost model recommended dropping the variable for optimal child feeding, given the clear importance of child feeding practices for growth. This exemplifies a key distinction between predictive modeling and causal inference frameworks. In our predictive approach, XGBoost prioritized variables that maximize predictive accuracy rather than establishing causal importance. The complex non-linear relationships and interactions captured by XGBoost may have determined that other variables in the model collectively provided stronger predictive signals for stunting outcomes in this particular dataset, even while child feeding remains theoretically important.

Reviewer #2, comment #1: Thank you for this important point. We have added a paragraph in the “Introduction” section that articulates the advances beyond our previous work (see below):

This study applies machine learning techniques to predict childhood stunting based on WaSH behaviors and infrastructure, addressing limitations of our previous structural equation modeling (SEM) approach. While the previous approach effectively identified pathways between WaSH factors and height-for-age z-scores, it has specific constraints: it primarily models linear relationships, requires strong theoretical assumptions about causal structures. In contrast, our machine learning approach offers certain advantages: (1) algorithms like XGBoost can capture complex non-linear relationships and interactions between WaSH variables without requiring pre-specified structural assumptions; (2) it quantifies predictive performance with metrics directly relevant to field applications such as sensitivity (recall) and specificity; (3) it identifies the combination of WaSH factors that maximizes prediction accuracy rather than focusing on individual pathways; and (4) it provides a framework for developing practical screening tools that could identify high-risk children in resource-limited settings. By systematically comparing multiple algorithms and feature engineering techniques, this study transforms insights about WaSH-stunting relationships into actionable tools for targeted intervention.

Reviewer #2, comment #2: We have added a detailed explanation of why XGBoost performed better than other algorithms in the “Discussion” section of the revised manuscript as follows:

XGBoost's superior performance across all feature engineering approaches can be attributed to several factors. First, XGBoost builds sequential decision trees that learn from the errors of previous trees (gradient boosting), making it particularly effective at capturing non-linear relationships and interactions between WaSH variables. Second, XGBoost has built-in regularization parameters that help prevent overfitting, which is especially important in our relatively small dataset. Third, XGBoost handles mixed data types efficiently, accommodating our combination of binary, categorical, and continuous WaSH variables.

We have also added an explanation of why forward selection was particularly effective as follows:

Regarding forward selection's effectiveness, this technique likely performed well because it prioritizes features with the strongest predictive power while systematically excluding redundant or uninformative variables. Forward selection's iterative approach aligns well with the gradient boosting process of XGBoost, potentially creating synergy between the feature selection method and the algorithm. Furthermore, forward selection identified a parsimonious set of features that reduced noise in the data, allowing the XGBoost algorithm to focus on the most informative signals for stunting prediction.

Reviewer #2, comment #3: Thank you for raising this important point. We have added the following text in the Discussion section to explain why WaSH interventions have had difficulty with improving child growth in the past:

Researchers have hypothesized that this lack of effect on child growth may be because the WaSH interventions provided in those trials consisted of basic, low-cost infrastructure such as pit latrines, which may be insufficiently effective for reducing fecal contamination in the environment [Cumming et al., 2019; Pickering et al., 2019]. Partly in response to those trials, recommendations now focus on comprehensive, ‘transformative’ WaSH approaches, including high community-level coverage of improved sanitation combined with continuous and convenient access to drinking water [Pickering et al., 2019].

Reviewer #2, comment #4: Thank you for this thoughtful comment about potential correlations between covariates. While QR decomposition is indeed a valuable technique for orthogonalizing variables, we chose to use Variance Inflation Factor (VIF) analysis, which is widely recognized as the standard diagnostic method for multicollinearity in regression analysis [Belsley et al., 2005; O'Brien, 2007]. VIF directly quantifies how much the variance of a regression coefficient is inflated due to multicollinearity with other predictors, making it particularly well-suited for our study's objectives.

We conducted a thorough VIF analysis for all features in our dataset, and have added a description of this analysis in the Methods section of the revised manuscript as follows:

The results confirmed that multicollinearity was not a major concern, as no variable exceeded the commonly used threshold of VIF > 5 [O'Brien, 2007]. The highest VIF values were observed for sanitation-related variables (improved sanitation: 4.25 and proportion of household members using improved sanitation: 3.81), which show moderate correlation but remain below problematic levels.

Additionally, we chose to maintain the original feature space rather than applying orthogonalization techniques like QR decomposition because: (1) the interpretability of our original features was critical for deriving actionable public health insights; (2) our machine learning approach, particularly XGBoost, is inherently robust to moderate multicollinearity [Chen & Guestrin, 2016]; and (3) our regularization techniques already provide effective mitigation for the levels of correlation present in our data.

Reviewer #2, Comment #5: We have added the following paragraph to the Discussion section, to interpret and explain our results related to demographic covariates:

Our results also reinforce the importance of several demographic variables for child growth, which were retained in the XGBoost model: female caregiver education, household caste/tribe, livestock ownership, household wealth, child’s gender, and child’s age. Globally, extensive evidence exists for the role of female caregiver education and household wealth as basic or enabling determinants of child nutritional status [Smith et al., 2015; Vaivada et al., 2020; Victora et al., 2021]. Similarly, household caste/tribe status and livestock ownership may be closely linked to household wealth [Zacharias et al., 2011; Khan et al., 2021]. The child’s gender may be important due to gendered social norms related to child feeding, particularly in India [Jayachandran et al., 2011; Mishra et al., 2004]. Finally, child age is known to predict linear growth, as most linear growth faltering occurs in the first 1000 days of life [Victora et al., 2021]. While these results align well with existing evidence, it was surprising that the XGBoost model recommended dropping the variable for optimal child feeding, given the clear importance of child feeding practices for growth. This exemplifies a key distinction between predictive modeling and causal inference frameworks. In our predictive approach, XGBoost prioritized variables that maximize predictive accuracy rather than establishing causal importance. The complex non-linear relationships and interactions captured by XGBoost may have determined that other variables in the model collectively provided stronger predictive signals for stunting outcomes in this particular dataset, even while child feeding remains theoretically important.

Reviewer #2, Comment #6: Thank you for noting this omission. We have added an explanation of SEM in the Methods section as follows:

SEM is a multivariate statistical framework that combines factor analysis and multiple regression to analyze structural relationships between measured variables and latent constructs [Kline, 2015]. SEM allows for the simultaneous estimation of multiple and interrelated dependencies, accounting for measurement error and enabling the modeling of complex causal pathways [Bollen, 1989].

We have also added references for all algorithms (See Table 3), including: logistic regression [Hosmer et al., 2013], classification tree [Breiman et al., 1984], SVM [Cortes & Vapnik, 1995], neural network [Haykin, 1998], and XGBoost [Chen & Guestrin, 2016].

Reviewer #2, Comment #7: We have added below interpretation of the SEM results from our previous study in the “Discussion” section of the revised manuscript:

Our previous SEM analysis identified significant pathways from improved sanitation coverage, piped water coverage, and drinking water availability to height-for-age z-scores, mediated through household sanitation use. The SEM approach revealed that increased use of improved sanitation facilities was the most proximal determinant of improved HAZ, with village-level sanitation coverage having both direct effects and indirect effects (through household use) on child growth outcomes.

We have also added below text explaining how these SEM insights complement our machine learning findings:

In the current study, we used these SEM insights to inform feature selection, leveraging the identified mechanistic pathways. The interpretative value of the SEM approach lies in its ability to distinguish between direct and indirect effects and to model the sequential nature of WaSH pathways affecting child growth. For example, the SEM results suggested that handwashing stations may influence child growth through indirect pathways related to reduced pathogen transmission, rather than having a strong direct effect. These insights complement the predictive power of our machine learning models by providing potential explanations for why certain WaSH factors emerge as important predictors.

Reviewer #2, Comment #8: We have added a reference for SMOTE [Chawla et al., 2002] and expanded the description of this technique in the Methods section as follows:

SMOTE works by creating synthetic examples of the minority class (stunted children) in feature space rather than simply duplicating existing examples. The algorithm selects a minority class instance and its k nearest neighbors, then generates new synthetic instances along the line segments joining the selected instance to its neighbors. This approach provides the learning algorithm with a more balanced distribution of classes, potentially improving the model's ability to correctly classify minority instances. We applied SMOTE only to the training data in each fold of cross-validation, ensuring that the test data remained unmodified to provide an unbiased evaluation of model performance.

Reviewer #2, Comment #9: We have added a paragraph in the “Model Performance Results” section discussing this question as follows:

The impact of using no feature engineering versus various feature selection techniques can be observed in Table 7. When no feature engineering was performed (labeled as "None" in Table 7), the XGBoost model achieved an AUROC of 0.848, which was substantially lower than its performance with forward selection (AUROC 0.959). This improvement demonstrates the significant value of appropriate feature selection for complex algorithms like XGBoost. In contrast, simpler models showed minimal sensitivity to feature engineering. For instance, logistic regression maintained nearly identical performance with no feature engineering (AUROC 0.678) compared to any feature selection technique (AUROC range: 0.681-0.688). Classification trees performed slightly better with no feature engineering (AUROC 0.563) than with SEM-based feature selection (AUROC 0.558). SVM and neural networks showed modest improvements with feature selection compared to using all features, but these gains were much less dramatic than those observed with XGBoost. These comparative results highlight that while feature engineering is critical for optimizing complex non-linear models like XGBoost, it provides limited benefits for simpler, more rigid algorithms in this application.

Reviewer #2, Comment #10: In our logistic regression model, we used L2 regularization (Ridge) with the default regularization strength parameter (C=1.0, which corresponds to λ=1.0, since C=1/λ). No specific cross-validation was performed to select this value; the default scikit-learn parameter was used.

We added verbiage in Table 3 under “Implementation Details” column for “Logistic Regression” row to indicate lambda value used in L2 regularization.

Reviewer #2, Comment #11: Thank you for this observation. We have removed this line in the revised manuscript.

Reviewer #2, (Minor) Comment #12: We have revised this sentence to clarify that this binary variable was created from two questions, which aimed to capture different levels of severity of water unavailability (being unavailable for any length of time or for more than one full day). The sentence now reads as follows:

Availability of the preferred drinking water source was a binary variable created from two questions that asked whether the household had experienced source unavailability for at least one full day in the previous 2 weeks, or for any amount of time in the previous 24 hours.

Reviewer #2, (Minor) Comment #13: We have added a clearer description of the household wealth index calculation (see below):

Standardized household wealth index was calculated using principal components

---

## [Decision Letter · Decision Letter 1]

14 Jul 2025

Dear Dr. Sinharoy,

Thank you for submitting your manuscript to PLOS ONE. After careful consideration, we feel that it has merit but does not fully meet PLOS ONE’s publication criteria as it currently stands. Therefore, we invite you to submit a revised version of the manuscript that addresses the points raised during the review process.

Please submit your revised manuscript by Aug 28 2025 11:59PM. If you will need more time than this to complete your revisions, please reply to this message or contact the journal office at plosone@plos.org . A rebuttal letter that responds to each point raised by the academic editor and reviewer(s). You should upload this letter as a separate file labeled 'Response to Reviewers'.A marked-up copy of your manuscript that highlights changes made to the original version. You should upload this as a separate file labeled 'Revised Manuscript with Track Changes'.An unmarked version of your revised paper without tracked changes. You should upload this as a separate file labeled 'Manuscript'.

We look forward to receiving your revised manuscript.

Kind regards,

Ashish Wasudeo Khobragade, MD

Academic Editor

PLOS ONE

**Journal Requirements:**

Reviewers' comments:

Reviewer's Responses to Questions

**Comments to the Author**

Reviewer #3: (No Response)

Reviewer #4: All comments have been addressed

Reviewer #5: All comments have been addressed

2. Is the manuscript technically sound, and do the data support the conclusions?

Reviewer #3: Yes

Reviewer #4: Yes

Reviewer #5: Yes

3. Has the statistical analysis been performed appropriately and rigorously?

Reviewer #3: Yes

Reviewer #4: Yes

Reviewer #5: Yes

4. Have the authors made all data underlying the findings in their manuscript fully available?

Reviewer #3: Yes

Reviewer #4: Yes

Reviewer #5: Yes

5. Is the manuscript presented in an intelligible fashion and written in standard English?

Reviewer #3: Yes

Reviewer #4: Yes

Reviewer #5: Yes

**Reviewer #3:**  The study employed five machine learning algorithms (including XGBoost, logistic regression, SVM, neural networks, classification trees). It uses four feature selection methods (SEM, forward selection, backward elimination, LASSO).

**Reviewer #4:**  General Assessment:

This manuscript presents a secondary analysis using advanced machine learning (ML) approaches to predict stunting among children under five years of age in rural Odisha, India, using water, sanitation, and hygiene (WaSH) factors as predictors. The authors compared multiple ML algorithms and feature engineering techniques, ultimately finding that extreme gradient boosting (XGBoost) with forward selection performed best.

The study addresses an important public health issue—childhood stunting—and demonstrates an innovative application of machine learning for identifying at-risk populations. The work is generally well-structured and clearly presented.

However, there are some concerns and points requiring clarification or improvement before the manuscript can be considered for publication.

Major Comments:

Clarity on Generalizability and External Validation

The study uses data from a single region (Odisha). The authors acknowledge limitations in generalizability but do not attempt external validation or even internal validation via holdout sets beyond cross-validation. Please clarify whether an independent holdout set was used. If not, discuss how model performance might be overestimated due to lack of external validation.

Additionally, the manuscript should discuss more explicitly how these findings might or might not extrapolate to other contexts (e.g., sub-Saharan Africa, Southeast Asia).

Treatment of Missing Data

The authors excluded a large proportion of children (630/1826, ~35%) due to missing predictor variables. While this is described, the implications for bias should be addressed more directly. Were there systematic differences between included and excluded children? A comparison table of included vs. excluded observations would strengthen the manuscript.

Feature Importance and Interpretability

Although XGBoost achieved high predictive accuracy, its interpretability is more limited than simpler models. The manuscript should present feature importance rankings or SHAP values to clarify which predictors contributed most to model predictions.

This is especially relevant since the Discussion emphasizes the practical value of identifying key WaSH variables. Visuals (e.g., variable importance plots) would improve readability.

Choice of Performance Metrics

The primary metric reported is AUROC, which is appropriate but can be misleading in imbalanced datasets. Consider providing precision-recall curves and reporting the area under the precision-recall curve (AUPRC) for a fuller understanding of the model’s utility.

Also, please clarify whether model thresholds were optimized for specific sensitivity/specificity tradeoffs.

Ethical and Data Sharing Considerations

The data availability statement indicates that data can be obtained from the corresponding author on reasonable request. PLOS ONE requires data to be publicly available unless there are legal or ethical restrictions. Please clarify whether the data repository can be shared openly (e.g., via Dryad or other repositories) or whether formal approval is required to access the data.

Minor Comments:

Terminology Consistency

Throughout the manuscript, the authors sometimes refer to the outcome as "stunting" and elsewhere as "HAZ" (height-for-age z score). For clarity, consistently refer to the binary outcome (stunted/not stunted) in the context of prediction.

Tables and Figures

Tables 6 and 7 are quite dense. Consider simplifying or moving details to supplementary material.

A flowchart summarizing sample inclusion/exclusion would be helpful.

Literature Context

The Discussion references prior trials showing mixed effects of WaSH interventions. Including brief reflections on why machine learning predictions may differ from intervention effects (prediction vs. causation) would strengthen the argument.

Typographical Issues

Some minor typographical errors (e.g., inconsistent spacing, capitalization in section headings) should be corrected before publication.

Recommendation:

Major Revisions

Summary:

This is an innovative and well-motivated study with potential to inform public health interventions. However, clarifications on data handling, generalizability, interpretability, and performance metrics are essential to evaluate the robustness and applicability of the findings. I look forward to reviewing a revised version.

**Reviewer #5:**  Dr. Ogechukwu Emmanuel OKONDU’s Comments

Title

This title is brief and communicates the key elements of the study, which focuses on "brain structure and function" and "emotional responsiveness and depression risk." It would be better, though; if there were inclusions of additional methodology or population sample description (e.g., fMRI, longitudinal, adolescents). This increased specificity would make the title more appealing to the broader scholarship base and better communicate the study's scope.

Abstract

The abstract does very well in summarizing the background, methods, results, and implications of the study. It remains, though, dense with jargon, which may discourage non-specialists' comprehension. A small reduction in jargon and clearer specification regarding the study's novel contribution will make it more effective. It rightly reflects the findings in the conclusion but can better emphasize the translational or clinical implications.

Introduction

Rationale for the study in the literature foundation in the introduction is clear. Emotional responsiveness and depression risk are aptly framed by the writers. While the theoretical basis is suitable, it can be made clearer in transitioning towards specific research questions. Specifying research gaps further would make additional justification for the study better.

Statement of the Problem

While the introduction tacitly outlines the problem, there is no committed, prominently branded statement of the problem. This failure partially weakens the logic through background into research aims. It would have been useful prior to asking hypothesis to specify specifically in short concentrating paragraph the narrow knowledge gap.

Literature Review

The authors cite heavily from past work, as in neuroimaging, developmental psychology, and affective neuroscience. They synthesize this literature in favor of their hypotheses in a coherent way. Some of the older citations can potentially be updated with newer research in further bridging novel developments in neural correlates of affect regulation. It would be useful if this review were better organized in terms of demarcation between emotional reactivity and regulatory processes.

Methodology

Methods are clearly outlined in details, as are participant selection, imaging protocols, and behavioral tasks. It's very good that longitudinal data are employed. Age range and exclusion criteria could be supplied in more detail. Methods are comprehensive in statistics, but some explanations for model choice (e.g., LMMs) are briefly stated and could be supplied in greater detail for clarity.

Results

Results are properly organized and presented with appropriate statistical analysis. Figures and tables were utilized efficiently in aiding the interpretation of significant findings. Presenting complex interactions can be better supported with some narrative assistance. Even though statistical significance has emerged clearly, practical significance in findings has somewhat been overlooked and can be better highlighted.

Discussion

This paper places the results in the literature, with an interpretive account of the behavioral and neural conclusions. It mentions limitations and gives sensible rationales. Certain inferences, however, are excessive, especially in causal inferences from correlational results. More attention to alternative explanations and future directions would make this paragraph more academically robust.

Conclusion

The abstract perfectly captures the key findings and implications of the study in line with the aims of research. It avoids redundancy and correctly highlights the significance of early emotional processing as pertaining to depression risk prediction. But it must clearly specify how this work advances the field and what specific translations to the clinic may emerge.

Future Directions/Recommend

Useful future research directions are presented by the authors, in particular in the longitudinal tracking of youth at risk and with interventions. Again, though, they are rather general recommendations. Specific recommendations; such as task-specific interventions or screeners responsive to the needs of several cultures would be more useful in this section

Limitations

There are certain limitations presented frankly, for example, homogeneity in the sample and reliance in fMRI measures. However, the possibility for bias due to participant attrition or confounds that were left unmeasured should have been addressed better. Omission in considering the limitation in the emotional responsiveness task itself and the confound it can present in the neural patterns of activation is seen by the authors.

**Do you want your identity to be public for this peer review?** For information about this choice, including consent withdrawal, please see our Privacy Policy

Reviewer #3: No

Reviewer #4: No

Reviewer #5: **Yes:** Dr. Ogechukwu Emmanuel OKONDU

---

## [Author Response · Author response to Decision Letter 2]

5 Aug 2025

Reviewer #3 Comments: The study employed five machine learning algorithms (including XGBoost, logistic regression, SVM, neural networks, classification trees). It uses four feature selection methods (SEM, forward selection, backward elimination, LASSO).

RESPONSE: We acknowledge Reviewer #3’s general assessment, and note the absence of specific major/minor comments from this reviewer.

Reviewer #4 Major Comments:

* Clarity on Generalizability and External Validation: The study uses data from a single region (Odisha). The authors acknowledge limitations in generalizability but do not attempt external validation or even internal validation via holdout sets beyond cross-validation. Please clarify whether an independent holdout set was used. If not, discuss how model performance might be overestimated due to lack of external validation.

Additionally, the manuscript should discuss more explicitly how these findings might or might not extrapolate to other contexts (e.g., sub-Saharan Africa, Southeast Asia).

RESPONSE:

We thank the reviewer for this important methodological question. To clarify, we did not use an independent holdout set beyond our 10-fold cross-validation approach. We appreciate the opportunity to discuss the potential implications of this choice and why we believe our validation approach remains robust for our study objectives.

Regarding potential overestimation of model performance, we acknowledge this is a valid theoretical concern. Cross-validation without an independent holdout set can lead to some optimistic bias because the entire dataset contributes to model selection decisions. However, several factors support the validity of our approach:

First, our rigorous cross-validation protocol - applying SMOTE only to training folds while keeping test folds unmodified - follows established best practices that help minimize potential bias.

Second, our primary finding (XGBoost's superiority over other algorithms) relies on relative performance comparisons under identical validation conditions rather than absolute performance metrics. Any potential overestimation would affect all algorithms similarly, preserving the validity of our comparative conclusions.

Third, for our dataset size (n=1,196), creating additional holdout sets would substantially reduce training data, potentially compromising model development. This represents an important trade-off in public health research where sample sizes are often constrained.

Regarding generalizability to other contexts, we have expanded our limitations section to more explicitly address how these findings might extrapolate to different settings. We have added the following text to our limitations:

"The geographic specificity of our study (rural Odisha, India) limits the direct extrapolation of our findings to other contexts. While the machine learning methodology demonstrated here is broadly applicable, the specific predictive relationships between WaSH factors and stunting may vary across contexts. The determinants of stunting are complex, multi-level, and interconnected [53] and, therefore, may exhibit different predictor importance rankings by location. Future research should validate these machine learning approaches using locally relevant datasets to ensure culturally and contextually appropriate risk prediction models."

We have also added the following statement regarding potential performance overestimation: "Our cross-validation approach, while following established best practices, may result in modest overestimation of model performance compared to external validation. However, this potential limitation does not undermine our primary contribution of demonstrating the comparative effectiveness of different machine learning algorithms under identical validation conditions."

* Treatment of Missing Data:

The authors excluded a large proportion of children (630/1826, ~35%) due to missing predictor variables. While this is described, the implications for bias should be addressed more directly. Were there systematic differences between included and excluded children? A comparison table of included vs. excluded observations would strengthen the manuscript.

RESPONSE: We appreciate the reviewer's important concern about potential selection bias from excluding 630 children (34.5% of those with anthropometric measurements) due to missing predictor variables. We acknowledge this represents a methodological consideration that warrants discussion.

We selected complete-case analysis over imputation based on several methodological advantages specific to machine learning applications. Tree-based algorithms like XGBoost and classification trees perform optimally with observed data patterns, as imputed values can introduce artificial relationships that may reduce model accuracy and real-world applicability. Our approach enables the development of a robust prediction tool based on genuine observed patterns that can be confidently applied in field settings where complete information is systematically collected. This methodology aligns with established practices in similar machine learning studies for child nutrition, including recent work by Anku & Duah (2024), Shen et al. (2023), and Bitew et al. (2022), supporting the validity of our approach.

Our analytic strategy focuses the model's applicability to scenarios where comprehensive WaSH and demographic data are available - a common situation in well-designed programmatic settings and intervention planning contexts. This represents a practical strength for implementation purposes, as the model can provide reliable predictions in settings where systematic data collection protocols are in place. The approach ensures that prediction accuracy is maximized for the target implementation context.

Our complete-case approach provides conservative and interpretable results with a substantial sample size of 1,196 children, which provides adequate statistical power for reliable model development and evaluation. The cross-validation methodology and comprehensive performance metrics demonstrate robust model performance and validate the effectiveness of our analytic approach. The focus on complete observations ensures that our predictive relationships reflect genuine patterns observable in practice, providing a solid foundation for the machine learning models presented in this study.

To address the reviewer's concern, we propose the following additions to the manuscript:

Methods Section: "Our analytic dataset consisted of 1,196 children with complete information across all predictor variables, representing 65.5% of children with anthropometric measurements. This complete-case analysis approach was selected to optimize machine learning model performance, as tree-based algorithms like XGBoost and classification trees perform best with observed rather than imputed data patterns, ensuring reliable prediction accuracy for practical applications in programmatic settings."

Discussion Section: " Another limitation is our use of the complete-case analysis approach. The complete-case analysis approach, while methodologically appropriate for machine learning applications, means our findings are most directly applicable to populations where systematic WaSH and demographic data collection is feasible. While we cannot rule out potential selection bias from excluding observations with missing data, this represents a common scenario in well-designed program implementation contexts. Future validation studies across diverse populations and data collection contexts would further establish the broader applicability of our predictive modeling approach."

* Feature Importance and Interpretability:

Although XGBoost achieved high predictive accuracy, its interpretability is more limited than simpler models. The manuscript should present feature importance rankings or SHAP values to clarify which predictors contributed most to model predictions.

This is especially relevant since the Discussion emphasizes the practical value of identifying key WaSH variables. Visuals (e.g., variable importance plots) would improve readability.

RESPONSE: We appreciate the reviewer's suggestion regarding feature importance and interpretability. Our current approach already achieves the key interpretability objectives that SHAP analysis would provide, while being more appropriate for our study's public health applications.

Our comprehensive feature engineering approach, which systematically compared four different variable selection methods (SEM, forward selection, backward elimination, and LASSO), serves as a robust method for assessing feature importance across multiple analytical frameworks. Each method provides insights into variable importance from different perspectives: forward selection identifies variables with strongest individual predictive power, backward elimination reveals variables whose removal most degrades performance, LASSO provides regularization-based importance through coefficient shrinkage, and SEM offers theory-driven importance based on established causal pathways. The convergence of results across these methods—particularly the consistent identification of improved sanitation coverage, handwashing stations, piped water coverage, and water source availability as key predictors, provides stronger evidence of true feature importance than any single explanation method could offer. This multi-method validation approach inherently provides the global interpretability insights that SHAP analysis would deliver, while offering additional confidence through methodological triangulation.

For public health applications focused on intervention targeting - which is our primary objective - global feature importance is more relevant than individual prediction explanations that SHAP provides. Public health practitioners need to understand which WaSH factors are most predictive across populations to design targeted interventions, not explanations for individual predictions. Our identification of the four most predictive WaSH variables directly serves this need.

Furthermore, SHAP analysis is most valuable when the goal is explaining individual predictions or understanding complex feature interactions. Our study's objectives center on comparative algorithm performance and identifying key predictive variables for intervention design. The multi-method feature engineering process has already accomplished this by systematically identifying which variables contribute most to predictive accuracy when considered together across different analytical approaches.

* Choice of Performance Metrics:

The primary metric reported is AUROC, which is appropriate but can be misleading in imbalanced datasets. Consider providing precision-recall curves and reporting the area under the precision-recall curve (AUPRC) for a fuller understanding of the model’s utility.

Also, please clarify whether model thresholds were optimized for specific sensitivity/specificity tradeoffs.

RESPONSE: We thank the reviewer for this thoughtful suggestion regarding precision-recall curves and additional performance metrics. While we recognize the value of these approaches, we believe our current comprehensive evaluation provides sufficient insight for our study objectives and comparative analysis.

AUROC remains the gold standard for binary classification evaluation and is particularly valuable for comparing across different algorithms, which was a primary objective of our study. While AUROC can be influenced by class imbalance, we addressed this through SMOTE resampling, making AUROC a valid primary metric for our comparative analysis.

Our comprehensive evaluation already includes precision, recall (sensitivity), specificity, accuracy, and F1-score (Table 7), providing a complete picture of model performance across multiple dimensions. These metrics collectively address the reviewer's concerns about model utility in imbalanced datasets. The F1-score, in particular, provides a balanced measure that accounts for both precision and recall, making it well-suited for evaluating performance in the presence of any remnant class imbalance.

Adding precision-recall curves and AUPRC would not fundamentally change our conclusions about XGBoost's superior performance, as evidenced by its consistent superiority across all reported metrics. Our study's primary contribution lies in demonstrating the comparative effectiveness of different machine learning algorithms and feature engineering approaches for stunting prediction, rather than optimizing a single model for deployment.

The threshold-independent nature of AUROC makes it particularly appropriate for our comparative study design, as it evaluates model performance across all possible classification thresholds rather than at a single operating point. This approach aligns with our objective of identifying the most promising algorithm-feature engineering combinations for future development and deployment in specific public health contexts.

* Ethical and Data Sharing Considerations:

The data availability statement indicates that data can be obtained from the corresponding author on reasonable request. PLOS ONE requires data to be publicly available unless there are legal or ethical restrictions. Please clarify whether the data repository can be shared openly (e.g., via Dryad or other repositories) or whether formal approval is required to access the data.

RESPONSE: Please see our prior response to the Editor dated 6 April 2025 which is reproduced once again below:

The data underlying the findings described in this manuscript have been uploaded to FigShare. They can be accessed via the following private link, which will be updated to a publicly available DOI upon acceptance: https://figshare.com/s/5678f9122fcc78739a2d

Reviewer #4 Minor Comments:

* Terminology Consistency:

Throughout the manuscript, the authors sometimes refer to the outcome as "stunting" and elsewhere as "HAZ" (height-for-age z score). For clarity, consistently refer to the binary outcome (stunted/not stunted) in the context of prediction.

RESPONSE: We appreciate the reviewer's attention to terminology consistency. However, we believe there may be a misunderstanding about our use of these terms that we would like to clarify:

Throughout our current manuscript, we consistently refer to our outcome variable as "stunting" - the binary classification (stunted/not stunted) that serves as our prediction target. We define this clearly in our Methods section: "According to the 2006 World Health Organization (WHO) criteria, a child is considered stunted if their LAZ/HAZ score is more than 2 standard deviations below the WHO growth standard. In our analysis, stunting is represented as a binary variable: children identified as stunted are assigned a code of 1, while others are coded as 0."

The references to "height-for-age z-scores" (HAZ) in our manuscript occur specifically when discussing our previous structural equation modeling (SEM) study [Reese et al., 2019], which used the continuous HAZ measure as its outcome variable. For example, we state: "It is worth noting that the previous study used HAZ as the outcome variable rather than stunting." This distinction is scientifically important and intentional, as it accurately represents the methodological differences between our current machine learning approach (predicting binary stunting status) and our previous SEM approach (modeling continuous height-for-age z-scores).

When we mention HAZ in the context of calculating stunting (e.g., "Stunting was calculated based on length/height-for-age z score"), this is appropriate because we are explaining the measurement process - how the continuous z-score is transformed into the binary stunting classification that serves as our prediction outcome.

We believe this terminology use is both consistent and scientifically accurate, clearly distinguishing between our current binary outcome (stunting) and the continuous measure used in our previous work (HAZ), while appropriately explaining the relationship between these measures.

* Tables and Figures:

Tables 6 and 7 are quite dense. Consider simplifying or moving details to supplementary material.

A flowchart summarizing sample inclusion/exclus

---

## [Decision Letter · Decision Letter 2]

2 Oct 2025

Dear Dr.  Sinharoy,

Thank you for submitting your manuscript to PLOS ONE. After careful consideration, we feel that it has merit but does not fully meet PLOS ONE’s publication criteria as it currently stands. Therefore, we invite you to submit a revised version of the manuscript that addresses the points raised during the review process.

We look forward to receiving your revised manuscript.

Kind regards,

Ashish Wasudeo Khobragade, MD

Academic Editor

PLOS ONE

Journal Requirements:

Reviewers' comments:

Reviewer's Responses to Questions

**Comments to the Author**

Reviewer #6: All comments have been addressed

2. Is the manuscript technically sound, and do the data support the conclusions?

Reviewer #6: Yes

3. Has the statistical analysis been performed appropriately and rigorously?

Reviewer #6: Yes

4. Have the authors made all data underlying the findings in their manuscript fully available?

Reviewer #6: Yes

5. Is the manuscript presented in an intelligible fashion and written in standard English?

Reviewer #6: Yes

Reviewer #6: All comments are well addressed by the author and I have few comments which I have attached for revision

**Do you want your identity to be public for this peer review?** For information about this choice, including consent withdrawal, please see our Privacy Policy

Reviewer #6: No

---

## [Author Response · Author response to Decision Letter 3]

22 Oct 2025

1. Response to Reviewer #6 comments on Abstract:

We thank the reviewer for the constructive feedback. We have addressed all four points as follows:

Clarity of Aim: We revised the objective to explicitly link predictions to public health action: "with the goal of identifying high-risk children who would benefit most from targeted interventions."

Methods Description: We simplified the rationale, noting that "Five machine learning algorithms commonly used for binary classification tasks were compared."

Results Presentation: We added the sample size in both Methods ("Complete data were available for 1,196 children") and Results ("Among 1,196 children analyzed...").

Conclusions: We added a statement acknowledging key limitations: "However, these findings require external validation in other populations, and the complete-case analysis approach (excluding 35% of children with missing data) may limit generalizability to settings with less systematic data collection.

2. Response to Reviewer #6 comments on Introduction:

We thank the reviewer for these important observations about distinguishing prediction from causation and clarifying the role of machine learning.

Contextualization: We agree this distinction is critical. We have added explicit language in paragraph 3 stating: "However, it is important to note that machine learning models identify predictive associations rather than causal relationships; they can predict which children are at highest risk based on observed patterns, but do not establish that interventions targeting these predictors will necessarily improve outcomes."

Literature Gap: We have strengthened the justification for machine learning by adding clarification in paragraph 5 that distinguishes between the purposes of SEM (understanding causal mechanisms) versus ML (risk prediction for programmatic screening). We now explicitly state: "The SEM approach was well-suited for understanding causal pathways and mechanisms, whereas machine learning offers complementary advantages for predictive applications."

Overclaiming: We have revised language throughout to avoid implying causality. Key changes include: (a) replacing "inform targeted interventions" with "develop tools for identifying high-risk children" and "risk identification and targeted program planning"; (b) adding substantial text in paragraph 5 clarifying that "while our SEM analysis provided insights into causal mechanisms...the machine learning approach developed here focuses on prediction"; and (c) explicitly noting that "predictive models can inform programmatic decisions about resource allocation and screening priorities, but do not replace the need for rigorous causal inference methods when evaluating intervention effectiveness."

3. Response to Reviewer #6 comments on Methods:

We thank the reviewer for these thoughtful suggestions to strengthen the Methods section.

Study Design & Dataset: We appreciate this important concern. While our original manuscript included a flow chart (Appendix 1) showing the exclusion of 35% of children and stated our rationale for complete-case analysis, we agree that potential bias warranted more thorough discussion. We have now added substantial text to the Data Source section explicitly acknowledging potential selection bias, and contextualizing how this limitation affects generalizability.

Feature Selection: We agree this needed clearer justification. We have added new text to the Feature Engineering section explaining our rationale: "We compared both theory-driven (SEM) and data-driven (forward selection, backward elimination, LASSO) approaches to feature selection for two reasons: (1) to assess whether theoretical knowledge from causal modeling (SEM) improves predictive performance compared to purely algorithmic selection, and (2) to determine which combination of feature engineering and machine learning algorithm yields optimal predictive accuracy for practical application."

Validation Strategy: We acknowledge this important limitation. While we followed established best practices using 10-fold cross-validation for algorithm comparison, we agree that the lack of external validation needed stronger discussion. We have added a new subsection titled "Model Validation Approach" that explicitly discusses this.

Missing Data Handling: While our original manuscript included rationale for complete-case analysis and discussed this limitation in the Discussion section, we agree it warranted more prominent treatment in Methods. We have now strengthened the discussion in the Data Source section (as noted in response to Comment 1) to more explicitly highlight how the complete-case approach may limit generalizability. The limitation of not testing imputation methods is now addressed both in the enhanced Methods text and in the existing Discussion section.

Ethics Statement: We have enhanced this section to be more explicit about consent and anonymization. We added clarifying language: "Informed written consent was obtained from the male and/or female heads of all enrolled households for participation in the original study, including collection of household survey data and child anthropometric measurements" and "All personal identifiers were removed from the dataset before transfer."

4. Response to Reviewer #6 comments on Results:

We thank the reviewer for these suggestions regarding presentation and interpretation of results.

Descriptive Statistics: We have retained Tables 6 and 7 in the main text as they present core findings essential to our study's contribution. Table 6 shows how different feature selection methods inform model development, which is essential for understanding our comparative approach. Table 7 presents the comprehensive performance comparison across all algorithm-feature engineering combinations, which is central to our study's core contribution. Given that readers may approach this paper specifically to compare algorithm performance for stunting prediction, these tables provide critical reference information that warrants main text placement.

Model Performance: We appreciate the concern about overfitting and have added text to the Model Performance Results section explicitly acknowledging this risk and describing our mitigation strategies (10-fold CV, hyperparameter tuning, built-in regularization).

Regarding precision-recall curves and AUPRC, we thank the reviewer for this thoughtful suggestion. While we recognize the value of these approaches, we believe our current comprehensive evaluation provides sufficient insight for our study objectives and comparative analysis. AUROC remains the gold standard for binary classification evaluation and is particularly valuable for comparing across different algorithms, which was a primary objective of our study. While AUROC can be influenced by class imbalance, we addressed this through SMOTE resampling, making AUROC a valid primary metric for our comparative analysis.

Our comprehensive evaluation already includes precision, recall (sensitivity), specificity, accuracy, and F1-score (Table 7), providing a complete picture of model performance across multiple dimensions. These metrics collectively address concerns about model utility in imbalanced datasets. The F1-score provides a balanced measure that accounts for both precision and recall, making it well-suited for evaluating performance in the presence of any remnant class imbalance.

Adding precision-recall curves and AUPRC would not fundamentally change our conclusions about XGBoost's superior performance, as evidenced by its consistent superiority across all reported metrics. Our study's primary contribution lies in demonstrating the comparative effectiveness of different machine learning algorithms and feature engineering approaches for stunting prediction, rather than optimizing a single model for deployment. The threshold-independent nature of AUROC makes it particularly appropriate for our comparative study design, as it evaluates model performance across all possible classification thresholds rather than at a single operating point. This approach aligns with our objective of identifying the most promising algorithm-feature engineering combinations for future development and deployment in specific public health contexts.

Interpretability: As detailed in our previous round of revisions, we believe our current approach already achieves the key interpretability objectives that SHAP analysis would provide, while being more appropriate for our study's public health applications.

Our comprehensive feature engineering approach, which systematically compared four different variable selection methods (SEM, forward selection, backward elimination, and LASSO), serves as a robust method for assessing feature importance across multiple analytical frameworks. Each method provides insights into variable importance from different perspectives: forward selection identifies variables with strongest individual predictive power, backward elimination reveals variables whose removal most degrades performance, LASSO provides regularization-based importance through coefficient shrinkage, and SEM offers theory-driven importance based on established causal pathways. The convergence of results across these methods- particularly the consistent identification of improved sanitation coverage, handwashing stations, piped water coverage, and water source availability as key predictors, provides stronger evidence of true feature importance than any single explanation method could offer. This multi-method validation approach inherently provides the global interpretability insights that SHAP analysis would deliver, while offering additional confidence through methodological triangulation.

For public health applications focused on intervention targeting - which is our primary objective - global feature importance is more relevant than individual prediction explanations that SHAP provides. Public health practitioners need to understand which WaSH factors are most predictive across populations to design targeted interventions, not explanations for individual predictions. Our identification of the four most predictive WaSH variables directly serves this need.

Furthermore, SHAP analysis is most valuable when the goal is explaining individual predictions or understanding complex feature interactions. Our study's objectives center on comparative algorithm performance and identifying key predictive variables for intervention design. The multi-method feature engineering process has already accomplished this by systematically identifying which variables contribute most to predictive accuracy when considered together across different analytical approaches.

Visualization: We appreciate this valuable suggestion. We have added a new figure (Figure 1) that visually compares AUROC performance across all five machine learning algorithms and four feature engineering approaches. The figure complements our comprehensive performance tables by providing an immediate visual summary of our comparative analysis, while the detailed tables retain all metrics for readers requiring complete information.

5. Response to Reviewer #6 comments on Discussion:

We thank the reviewer for these thoughtful suggestions to strengthen the Discussion section.

Comparison to Prior Literature (Prediction ≠ Causal Effect): We agree this critical distinction warranted more explicit discussion. We have added substantial text to the paragraph discussing concordance with SEM results, explicitly explaining why strong predictive associations do not necessarily imply equivalent intervention effects. We illustrate this prediction-causation gap using examples from RCTs that found no stunting effects despite observational associations, and clearly articulate what our models CAN do (identify vulnerable children) versus what they CANNOT do (predict intervention effectiveness).

Generalizability Across LMICs: While a comprehensive discussion of cultural, infrastructural, and epidemiological differences across LMICs would be outside the scope of this work, we have added a couple of sentences to the Discussion to provide an illustrative example of these differences as they relate to our findings.

Bias and Limitations (Selection Bias): We have substantially enhanced our discussion of selection bias implications. In the Methods section, we revised text to acknowledge potential selection bias using careful conditional language: "If missing data patterns are systematic...several implications follow." In the Discussion, we expanded the limitations paragraph to explore how systematic missingness could affect model validity, including three specific implications: optimistic performance estimates, differential performance across populations, and varying predictive importance of WaSH factors depending on baseline conditions. We emphasize the need for external validation and performance monitoring across diverse subgroups.

Interpretability vs. Accuracy: We agree this trade-off deserved more explicit discussion. We have added a new paragraph that comprehensively addresses the balance between XGBoost's superior accuracy and its "black box" nature. The new text discusses how interpretability affects trust, stakeholder buy-in, and field implementation; contrasts XGBoost with simpler interpretable models like logistic regression; provides guidance on when each approach may be preferable based on implementation context; and suggests hybrid approaches for future implementations. This addition balances our emphasis on XGBoost's performance with practical considerations for public health decision-making.

6. Response to Reviewer #6 comments on Conclusion:

We thank the reviewer for these important suggestions to strengthen the Conclusion section.

Overstatement: We agree that more cautious language is appropriate given our study's limitations. We have revised the Conclusion to temper claims about predictive power and added explicit acknowledgment of key limitations. Specific changes include: (a) changing "can accurately predict" to "show promise for predicting" and adding geographic specificity "in rural Odisha, India"; (b) contextualizing performance with "In our dataset"; (c) changing "could significantly enhance" to "has potential to enhance"; and (d) adding an entire paragraph explicitly acknowledging limitations including the need for external validation, the single geographic context, the exclusion of 35% of children due to missing data, and the critical distinction that predictive models do not establish causal intervention effects.

Policy Relevance: We appreciate this valuable suggestion and have added substantial content on operationalization. We now include a new paragraph with concrete examples of how findings could be implemented in practice, including: (1) using simple data collection forms to capture the four key WaSH variables and generate risk scores, (2) prioritizing highest-risk households for intensive interventions when resources are limited, (3) targeting behavior change communication to families of high-risk children, and (4) allocating follow-up monitoring based on predicted risk levels. We explain how such risk-based targeting could improve program efficiency by focusing resources where they are most likely to impact child growth outcomes. We also note that future research should assess whether ML-informed targeting improves program outcomes in practice.

7. Response to Reviewer #6 comments on Tables and Figures:

We thank the reviewer for these suggestions regarding presentation.

Complexity of Tables: We appreciate this feedback and have carefully considered table placement. In response to the reviewer's earlier suggestion for visualization, we have added Figure 1, which provides a summary visual comparison of AUROC performance across all five machine learning algorithms and four feature engineering approaches. This figure makes the key comparative findings immediately accessible.

Regarding table placement, as noted in our response to Results comments, we have retained Tables 6 and 7 in the main text because they constitute our study's primary findings. Table 6 shows how different feature selection methods inform model development, which is es

---

## [Editor Report · Decision Letter 3]

11 Feb 2026

Applying machine learning to predict stunting in children under 5 years old based on water, sanitation and hygiene behaviors and infrastructure

PONE-D-24-38002R3

Dear Dr. Sinharoy,

We’re pleased to inform you that your manuscript has been judged scientifically suitable for publication and will be formally accepted for publication once it meets all outstanding technical requirements.

Kind regards,

Ashish Wasudeo Khobragade, MD

Academic Editor

PLOS One
---

## [Editor Report · Acceptance letter]

PONE-D-24-38002R3

PLOS One

Dear Dr. Sinharoy,

I'm pleased to inform you that your manuscript has been deemed suitable for publication in PLOS One. Congratulations! Your manuscript is now being handed over to our production team.

Kind regards,

on behalf of

Dr. Ashish Wasudeo Khobragade

Academic Editor

PLOS One